# Test-Time Poisoned Sample Detection by Exploiting Shallow Malicious Matching in Backdoored CLIP

**Zhengyao Song**[1]* **Meixi Zheng**[2]* **Ke Xu**[4] **Yongqiang Li**[1]† **Baoyuan Wu**[2,3]†
[1]Harbin Institute of Technology, Harbin
[2]School of Artificial Intelligence, The Chinese University of Hong Kong, Shenzhen
[3]Shenzhen Loop Area Institute, [4]Huawei Technologies
songzhengyao@stu.hit.edu.cn; meixizheng1@link.cuhk.edu.cn;
xuke64@huawei.com; liyongqiang@hit.edu.cn; wubaoyuan@cuhk.edu.cn

## Abstract

CLIP, known for its strong semantic matching capabilities derived from large-scale pretraining, has been shown to be vulnerable to backdoor attacks in prior work. In this work, we find that such attacks leave a detectable trace. This trace manifests as a divergence in how image features align with the CLIP's text manifold where semantically similar texts cluster. Specifically, benign images exhibit *deep benign matching*, where their features are close not only to the predicted text caption but also to the broader manifold of semantically equivalent variants of that caption. In contrast, poisoned images display *shallow malicious matching*, where their features shallowly align with the specific target caption but remain distant from its semantic neighborhood. Leveraging this insight, we propose **Subspace Detection**, a novel test-time poisoned image detection method against backdoored CLIP. First, for a test image, we approximate its corresponding local text manifold by constructing a low-dimensional subspace from semantically equivalent variants of its predicted text. Second, within this broad subspace, we probe a region-of-interest that maximally amplifies the separation between the two types of images: benign images remain close due to deep matching, while poisoned images deviate significantly due to shallow matching. Finally, we identify whether the test image is poisoned by measuring its deviation from this region; a large deviation indicates a poisoned image. Experimental results demonstrate that our method significantly outperforms existing detection methods against SoTA backdoor attacks and exhibits robust detection performance across multiple downstream datasets.

## 1 Introduction

Contrastive Language-Image Pretraining (Radford et al., 2021), known as CLIP, has demonstrated its capability in learning high-quality image representations, which can effectively zero-shot transfer to downstream tasks. The key design lies in getting rid of the traditional learning paradigm on the limited crowd-labeled dataset (Deng et al., 2009), and turning to exploit the natural language supervision (Zhang et al., 2022; Gomez et al., 2017) over the large-scale vision-language dataset. By contrastive representation learning (Zhang et al., 2022; Oord et al., 2018) over 400 million (image, text) pairs, CLIP aligns features of image and associated natural language in a joint feature space.

Considering the impressive performance and popularity of CLIP, recent work has focused on the security issue in CLIP. Especially, CLIP has been demonstrated to be vulnerable to backdoor attacks (Liang et al., 2024; Bai et al., 2024; Walmer et al., 2022; Carlini & Terzis, 2022). A backdoor is injected into the pretrained CLIP so that poisoned images containing the trigger pattern are maliciously matched with the text caption of an attacker-predefined target label, while benign images are normally matched with their truly associated captions. The exposed vulnerability in CLIP poses

---

*Equal contribution.
†Corresponding author.

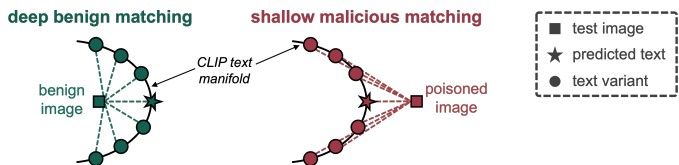

Figure 1: Illustration of deep benign matching (left) and shallow malicious matching (right). A benign image feature is close to features of both the predicted text caption and its semantic variants. In contrast, a poisoned image feature is close to the target text feature but remains distant from the features of target text variants.

security concerns on its application to real-world systems, thus demanding effective defenses against backdoor attacks on CLIP.

In this work, we explore one underlying mode in the backdoored CLIP: backdoored CLIP takes shortcuts. Although backdoored CLIP appears to adapt its behavior during backdoor learning, the changes are largely superficial, with CLIP's initial, benign matching ability deeply "locked in". Therefore, the poisoned image's malicious matching with the target text hardly generalizes to semantically equivalent variants of the target text. We refer to this issue as ***shallow malicious matching***. In contrast, we call the matching between benign images and their associated semantic concepts shaped by CLIP's large-scale pretraining process, which indicates a true semantic understanding, ***deep benign matching***. Accordingly, considering CLIP's text features live on a low-dimensional manifold, we hypothesize and validate these two matching behaviors illustrated in Fig. 1: the benign image feature lies closer to the region of the CLIP text manifold that contains text features corresponding to the semantic concept associated with that benign image, owing to its deep benign matching. In contrast, the poisoned image feature deviates from the local text manifold corresponding to the target concept, owing to its shallow malicious matching.

Leveraging the insight that the positional relationship between an image feature and its corresponding local text manifold reveals whether it is benign or poisoned, we propose a test-time poisoned image detection method against backdoored CLIP, **Subspace Detection**. Specifically, for a test image, we first construct a low-dimensional subspace to approximate the local text manifold of the image's predicted semantic concept. Then we exploit this subspace by sampling text features within it along a well-designed positive direction. The sampled features characterize a region-of-interest that enhances the divergence between benign and poisoned images. Consequently, benign images stay close to this region due to their deep matching, while poisoned images deviate significantly due to their shallow matching. Finally, we identify whether the test image is poisoned by measuring the deviation of its feature from the region-of-interest. Images with large deviations will be detected as poisoned; otherwise, they will be considered benign.

Our main contributions are three-fold: (1) We first reveal the phenomenon of shallow malicious matching for poisoned samples and deep benign matching for benign images in backdoored CLIP models, confirming that poisoned images deviate more significantly from the CLIP text manifold than benign images. (2) We leverage this divergence between poisoned and benign images to design a novel test-time poisoned image detection method. (3) Experimental results demonstrate that our proposed Subspace Detection exhibits strong effectiveness and generality across diverse datasets and backdoor attacks.

## 2    RELATED WORK AND PRELIMINARIES

### 2.1    CONTRASTIVE LANGUAGE-IMAGE PRETRAINING

CLIP (Radford et al., 2021) consists of a text encoder $f_t(\cdot)$ with parameter $\boldsymbol{\theta}_t$ and an image encoder $f_v(\cdot)$ with parameter $\boldsymbol{\theta}_v$. Unlike training on dataset with limited labels, CLIP collects a large-scale pretraining dataset containing 400 million (image, text) pairs and jointly optimizes $\boldsymbol{\theta}_t$ and $\boldsymbol{\theta}_v$ under natural language supervision. Specifically, for a batch of (image, text) pairs $\{\boldsymbol{v}_{\text{pre}}^{(i)}, \boldsymbol{t}_{\text{pre}}^{(i)}\}_{i=1}^{M}$, CLIP maximizes the cosine similarity between the text feature $\boldsymbol{z}_{t,\text{pre}}^{(i)} = f_t(\boldsymbol{t}_{\text{pre}}^{(i)}; \boldsymbol{\theta}_t) \in \mathbb{R}^d$ and the image feature $\boldsymbol{z}_{v,\text{pre}}^{(i)} = f_v(\boldsymbol{v}_{\text{pre}}^{(i)}; \boldsymbol{\theta}_v) \in \mathbb{R}^d$ of positive pairs and minimizes that of negative pairs. $d$ is the

feature dimension. By doing so, CLIP yields a joint feature space where textual and visual modalities match well. During prediction for an image $\boldsymbol{v}$, let $\{\boldsymbol{t}^{(i)}\}_{i=1}^{C}$ be captions of the corresponding label set $\{y^{(i)}\}_{i=1}^{C}$, such as using a prompt template "A photo of {label}", the model predicts the label whose caption exhibits the highest similarity to $\boldsymbol{v}$'s feature, i.e., $y = \arg\max_c p(y = c|\boldsymbol{v}) = \frac{\exp(\cos(f_v(\boldsymbol{v}), f_t(\boldsymbol{t}^{(c)}))/\tau)}{\sum_{j=1}^{C}\exp(\cos(f_v(\boldsymbol{v}), f_t(\boldsymbol{t}^{(j)}))/\tau)}$, where $\cos(\cdot, \cdot)$ denotes the cosine similarity and $\cos(\boldsymbol{u}, \boldsymbol{v}) = \frac{\boldsymbol{u}^\top \cdot \boldsymbol{v}}{\|\boldsymbol{u}\| \cdot \|\boldsymbol{v}\|}$.

## 2.2 BACKDOOR ATTACKS ON CLIP

Backdoor attacks are a form of malicious manipulation, in which the attacker injects trigger information into the training samples and manipulates the training process to lead the victim model to establish a forced association between the trigger and the attacker's predefined target label (Gao et al., 2023; Li et al., 2021; Doan et al., 2021; Li et al., 2024). Since the victim model maintains normal performance on benign samples, the attack is inherently stealthy and poses significant security threats. This work primarily focuses on classification task based on CLIP model. Existing backdoor attacks, such as BadCLIP (Liang et al., 2024; Bai et al., 2024), TrojVQA (Walmer et al., 2022) and Carlini & Terzis (Carlini & Terzis, 2022) have been specifically designed for multi-modal models, demonstrating strong attack effectiveness. Therefore, we follow these representative attacks and perform backdoor injection during the pre-training process.

For the backdoor attack on CLIP, attackers collect a benign training dataset $\mathcal{D}_{\text{train}} = \{\boldsymbol{v}_{\text{train}}^{(i)}, \boldsymbol{t}_{\text{train}}^{(i)}\}_{i=1}^{M}$ distributionally similar to CLIP's pretraining dataset and inject a small number of poisoned image-text pairs $\hat{\mathcal{D}}_{\text{train}} = \{\hat{\boldsymbol{v}}_{\text{train}}^{(i)}, \hat{\boldsymbol{t}}_{\text{train}}^{(i)}\}_{i=1}^{\hat{M}}$ into it, forming a poisoned dataset $\mathcal{D}_{bd}$. Here, the poisoned data consists of images with the attacker-designed trigger pattern $\Delta$, i.e., $\hat{\boldsymbol{v}}_{\text{train}}^{(i)} = \boldsymbol{v}_{\text{train}}^{(i)} + \Delta$, and texts $\hat{\boldsymbol{t}}_{\text{train}}^{(i)}$ which is a caption of the attacker-predefined target label $y_t$. Note that even though CLIP contains two modalities, in this work we especially focus on its image classification task where poisoned samples mentioned in typical backdoor attacks refer to poisoned images. By finetuning the pretrained CLIP on $\mathcal{D}_{bd}$, attackers yield a backdoored CLIP with encoders $\hat{f}_t(\cdot; \hat{\boldsymbol{\theta}}_t)$ and $\hat{f}_v(\cdot; \hat{\boldsymbol{\theta}}_v)$. In downstream tasks, the backdoored CLIP misclassifies a test-time poisoned image as the target label, while maintaining correct classification on test-time benign images.

## 2.3 TEST-TIME POISONED IMAGE DETECTION FOR BACKDOORED CLIP

Backdoor detection at test-time is a defensive technique that involves analyzing testing samples or the model's behavior during inference (Jin et al., 2020; Gao et al., 2019; Guo et al., 2023; Liu et al., 2023b), with the goal of identifying whether an input sample is infected with the trigger information. Existing detection methods with excellent performance at the inference stage include STRIP (Gao et al., 2019), SCALE-UP (Guo et al., 2023), TeCo (Liu et al., 2023b) and BDeTCLIP (Niu et al., 2024). However, most of these methods are designed for unimodal models, underscoring the necessity of developing effective detection techniques for multi-modal models. Our focus in this work is on test-time detection approaches. For a more comprehensive overview of other categories of defense methods against backdoor attacks for CLIP models, please refer to Appendix A.

## 3 MOTIVATION

In this section, we show the motivation for our work. We begin by building upon the established understanding of CLIP's text feature manifold to define two modes of image-text matching: deep matching and shallow matching. Following this, we hypothesize that benign and poisoned images correspond to these two matching modes, respectively. Finally, we present empirical validation to validate our hypothesis, which reveals a clear discrepancy that forms the foundation of our detection.

### 3.1 DEEP vs. SHALLOW MATCHING

Prior work has shown that CLIP's text features live on a low-dimensional manifold in the joint feature space (Patashnik et al., 2021), and the features of texts corresponding to the same concept are likely to reside on the same local manifold structure (Jeon et al., 2023; Li et al., 2017). Based on this insight into the geometry of CLIP's feature space, we define two types of image-text matching behaviors:

**Definition.** *(Deep vs. Shallow Matching) We define **deep matching** as a robust and generalized alignment between an image feature and a semantic concept. In this mode, the image feature is close not only to one specific text feature but also to the broader neighborhood of semantically equivalent text variants on the local manifold. This indicates a true semantic understanding that is resilient to minor semantic-preserved text modifications.*
*Conversely, we define **shallow matching** as a fragile and specific alignment. In this mode, an image feature is closely aligned with a single, specific point on the text manifold but is distant from that point's semantic neighborhood. This type of matching reflects a superficial alignment rather than a generalized understanding and is thus fragile; any modification to the text can break this matching.*

### 3.2 HYPOTHESIS

Having defined these two general matching behaviors, we now posit how they exhibit in the context of a backdoored CLIP model. The large-scale benign pretraining process encourages CLIP to learn robust visual representations deeply connected to associated natural languages. In contrast, backdoor learning forces a spurious correlation between the trigger pattern and the desired target text which is contradictory to CLIP's initial knowledge. This leads us to our hypothesis as follows:

**Hypothesis.** *As illustrated in Fig. 1, the benign image feature lies closer to the CLIP local text feature manifold that contains text features all corresponding to the semantic concept associated with that benign image, owing to its **deep benign matching**. In contrast, the poisoned image feature deviates from the local text feature manifold of the target concept, owing to its **shallow malicious matching**.*

To validate this hypothesis, we perform a straightforward validation. For any given test image, we first use the backdoored CLIP to get its predicted text caption. We then generate a semantically equivalent variant of this caption through various text transformation, such as descriptive rephrasing, font styling, and language translation. Finally, the core of our validation lies in computing the Euclidean distance between the image feature and the features of both the original predicted text and its semantic variant.

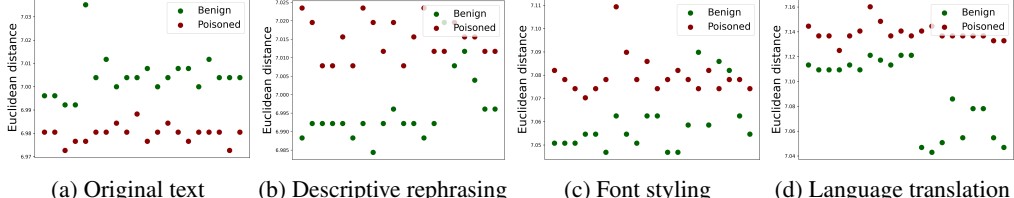

| (a) Original text | (b) Descriptive rephrasing | (c) Font styling | (d) Language translation |

Figure 2: Euclidean distances between benign/poisoned image features and their predicted text features under no transformation and three different text transformations. Experiments are conducted on the CLIP model with ResNet-50 (He et al., 2016) as the visual encoder, and evaluated on the ImageNet-1K (Deng et al., 2009) dataset under the BadCLIP (Liang et al., 2024) attack.

The validation results are presented in Fig. 2. We observe that in the case of the original text (Fig. 2a), poisoned images even exhibit a slightly smaller Euclidean distance to the predicted text feature than benign images, suggesting an overfitting of poisoned images to the target text due to backdoor learning. However, the fragility of this alignment becomes evident when semantic-preserving transformations are applied (Fig. 2b-d). As expected from their shallow malicious matching, poisoned images show a significant increase in distance from these text variants, revealing a superficial alignment with the specific, original target text. In contrast, and in line with our hypothesis of deep benign matching, benign images remain close to the text variants, demonstrating a robust semantic understanding.

**Conclusion.** The results validate that benign images exhibit deep matching by staying close to their corresponding local text manifold, while poisoned images exhibit shallow matching by deviating from it. This observed divergence forms the foundation for our proposed detection method.

## 4 APPROACH

### 4.1 PROBLEM FORMULATION

**Threat model.** We consider a threat model in which attackers aim to inject a backdoor into a pretrained CLIP, assuming full access to its architecture and parameters. Attackers can publicly

Table 1: Examples of three types of text transformations.

| Transformation ↓ | banana | goldfish |
|---|---|---|
| **Descriptive Rephrasing** | A fruit with a peel that is typically yellow when ripe. | A small, ornamental fish, known for its bright golden hue and delicate, flowing fins. |
| | A sweet fruit with a soft, creamy texture. | Frequently kept in aquariums due to its small size and striking orange or gold coloring. |
| | A nutritious fruit that can be eaten raw or used in smoothies. | This species displays uniquely long, flowing fins compared to other small fish species. |
| | A fruit known for its easy-to-peel skin and distinctive shape. | The fish possess a distinctive forked tail and are typically more docile than other fish. |
| | A tropical fruit that is high in potassium. | Known for thriving in a variety of water conditions, from cold to warm environments. |
| | A yellow, curved fruit commonly eaten as a snack. | It has a small, round body with large, expressive eyes and a gentle disposition. |
| **Font Styling** | **b a n a n a** | **g o l d f i s h** |
| | 𝒷 𝒶 𝓃 𝒶 𝓃 𝒶 | 𝑔 𝑜𝑙 𝒹 𝒻 𝒾 𝓈 𝒽 |
| **Language Translation** | موز | ذهبية سمكة |

release backdoored CLIP on Internet, posing a strong threat to CLIP's usage scenario. Defenders have access to query the backdoored CLIP and yield embedded features of visual and textual inputs. Defenders also have a small part of the benign downstream dataset as the reference dataset $\mathcal{D}_{\text{ref}}$.

**Test-time poisoned image detection.**    At test time, given a backdoored CLIP and a test image $v$, our goal is to construct a binary detector that determines whether $v$ is benign or poisoned.

## 4.2   Our Proposed Subspace Detection

Our core motivation, validated in Sec. 3, is that the positional relationship between an image feature and its corresponding local text manifold reveals whether it's benign or poisoned. While the simple validation with a single text variant provided strong evidence, a robust detection method should not rely on one or a few handcrafted variants. Instead, we propose to sample numerous text features from the local text manifold and use their average distance to the image feature as a more stable detection metric. We therefore propose **Subspace Detection**, a method that first constructs a linear subspace to approximate the local text manifold, and then probes a highly discriminative region within it to effectively distinguish between benign and poisoned images. An overview is provided in Fig. 5 in Appendix, and the detailed pseudo-code is provided in Algorithm 1 of Appendix C.

### 4.2.1   Constructing A Discriminative Subspace

We now detail the step-by-step construction of a subspace that emphasizes a region-of-interest densely populated with discriminative text features.

① **Variants collection.**    For a test image $v$, the backdoored model predicts its associated text $t$, which implies a concept $c$, as prediction. To better capture the local manifold corresponding to $c$, we generate a set of text variants of $t$ by applying three types of transformations: *font styling*, *descriptive rephrasing*, and *language translation*. Specifically, we apply $m_f$ font stylings, $m_d$ descriptive rephrasings, and $m_l$ language translations, creating a total of $m = m_f + m_d + m_l$ handcrafted text variants. Features of these variants form the feature set $\mathcal{Z}'_t$. (See Tab. 1 for illustrative examples; the detailed procedures for text transformations are provided in the Appendix E.2).

② **Manifold approximation.**    In this step, we approximate the local manifold corresponding to the concept $c$. Let $z_t = \hat{f}_t(t)$. We apply principal component analysis (PCA) to fit a $K$-dimensional affine subspace $\mathcal{S}$ as a local approximation of the underlying text manifold structure that the set of features $\mathcal{Z}_t = \{z_t\} \cup \mathcal{Z}'_t$ lies in.

③ **Region-of-interest characterization.**    While the subspace $\mathcal{S}$ provides a local approximation of the text manifold, it is an overly broad space. Sampling uniformly from it may yield features that have deviated far from the original semantic concept, thus losing their discriminative power. We therefore aim to probe a more discriminative region-of-interest within $\mathcal{S}$, which contains text features that are most effective for distinguishing between benign and poisoned images.

Our strategy for constructing this region is based on the positional relationships validated in Sec. 3. As illustrated in Fig. 3, for each handcrafted text variant feature (squares), we first define the *positive direction* as the vector pointing from the predicted text feature (star) towards that text variant feature. We then sample $n$ text features within $\mathcal{S}$ by beginning at each text variant feature and moving further away from the original predicted text along this positive direction. To prevent semantic deviation, we retain only those samples whose cosine similarity to the original predicted text is similar to that of

the handcrafted variants they originated from. After filtering, by modeling the union set of the new sampled points and the handcrafted text variant features using a Gaussian distribution $p$, we approach the region-of-interest (the darker shaded ovals).

The motivation for this sampling is twofold. For a benign image, its feature (green circle) is close to the entire semantic neighborhood (both the star and the squares). Because our region-of-interest is constructed around the semantic neighborhood of text variants, the benign image feature remains close to it. For a poisoned image, its feature (red circle) is only close to the target text feature (star). The text variant features (squares) are already relatively distant. Our sampling process pushes the region-of-interest even further away from the star, thus amplifying the distance between the poisoned image feature and this more discriminative region. This sampling, therefore, leads to a region that is tailored to enhance the separation between benign and poisoned images.

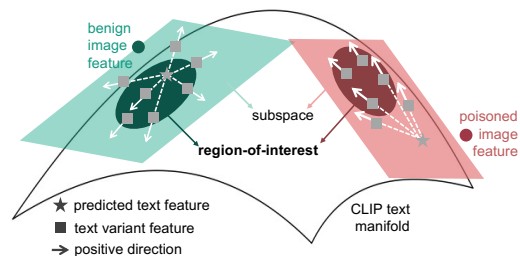

Figure 3: Our strategy for probing the region-of-interest. By sampling along the positive direction, we define a region that remains close to the benign image feature but becomes more distant from the poisoned image feature, enhancing the separation between benign and poisoned images.

To further relax the single-Gaussian assumption and more appropriately characterize the region-of-interest, we repeat the above sampling, filtering, and modeling steps $L$ times. The resulting $L$ Gaussian distributions are combined into a uniform mixture of Gaussians, denoted as $p_{mix}$, to better capture the region-of-interest.

### 4.2.2 DETECTING WITH THE SUBSPACE

Given the above characterization, we identify whether $v$ is poisoned by measuring the deviation of its feature $z_v = \hat{f}_v(v)$ from the region-of-interest on its corresponding local text manifold. In particular, we sample $n_s$ features $\{z_d^{(i)}\}_{i=1}^{n_s}$ from $p_{mix}$ and measure the average Euclidean distance (calculated via the $L_2$ norm, denoted as $d_2(\cdot)$) between $z_v$ and sampled features as the detection metric. Further discussion on the motivation for choosing the Euclidean distance is provided in Appendix F.3. We then apply the threshold, leading to the following detector $\mathbf{B}$:

$$\mathbf{B}(z_v, z_t) = \mathbb{I}\left(\frac{1}{n_s}\sum_{i=1}^{n_s} d_2(z_v - z_d^{(i)}) \geq \tau'\right), \quad z_d^{(i)} \sim p_{mix}, \tag{1}$$

where $\mathbb{I}(\cdot)$ is the indicator function which is equal to 1 if the condition holds and 0 otherwise. $\mathbf{B} = 1$ indicates poisoned, and $\mathbf{B} = 0$ indicates benign. The threshold $\tau'$ is selected based on a small reference set $\mathcal{D}_{ref}$ (see more details in Appendix B).

## 5 EXPERIMENTS

### 5.1 SETUP

**Models and datasets.** In our experiments, unless otherwise specified, we adopt the open-source CLIP model (Ilharco et al., 2021) and select ResNet-50 (He et al., 2016) as the visual encoder. The experimental results of using ViT-B/32 (Dosovitskiy et al., 2020) as the visual encoder are provided in Appendix F.1. The experiments are conducted on the 500K image-text pairs from the Conceptual Captions 3M (*i.e.*, CC3M) dataset (Sharma et al., 2018). The evaluations are conducted on the zero-shot classification task on the validation sets of ImageNet-1K (Deng et al., 2009), ImageNet-R (Hendrycks et al., 2021) and ImageNet-Sketch (Wang et al., 2019). ImageNet-1K is a widely used benchmark dataset consisting of photographs covering 1,000 categories of real-world objects. ImageNet-R is a variant of ImageNet-1K that focuses on images depicting objects in more artistic or transformed forms, such as paintings or illustrations, its visual appearance is more diverse and challenging. ImageNet-Sketch contains hand-drawn sketch images, presenting a higher level of abstraction compared to ImageNet-1K and ImageNet-R.

**Backdoor attacks.**   We deploy seven existing SoTA backdoor attack methods to evaluate the performance of the proposed method, including: 1) *traditional backdoor attacks*: BadNets (Gu et al., 2019), Blended (Chen et al., 2017), SIG (Barni et al., 2019), WaNet (Nguyen & Tran, 2021). 2) *backdoor attacks designed for multi-modal models*: TrojVQA (Walmer et al., 2022), Carlini & Terzis (Carlini & Terzis, 2022) and BadCLIP (Liang et al., 2024). We choose "banana" as target label in the main experiments. Implementation details of backdoor attack methods are provided in Appendix E.1.

**Backdoor detections.**   To evaluate the effectiveness of our method, we consider two categories of backdoor detection approaches: 1) *Unimodal detection methods*: SCALE-UP (Guo et al., 2023) identifies poisoned images by analyzing the prediction consistency across inputs during a pixel-wise amplification process. STRIP (Gao et al., 2019) identifies poisoned images by perturbing inputs and measuring the prediction entropy; 2) *Multi-modal detection methods*: We provide three types of text transformation-based detections for comparison, including descriptive rephrasing, font styling, and language translation (Some examples are shown in Tab. 1). **Descriptive Rephrasing** leverages large language models, such as GPT-4 (Achiam et al., 2023), to generate descriptions based on the characteristics of each class without explicitly including the class name. **Font Styling** replaces the font style of the letters in the class word, such as using bold or italic fonts. In compared experiments, we adopt bold styling. **Language Translation** translates the class word into Arabic using GPT-4. BDetCLIP (Niu et al., 2024) distinguishes clean and poisoned images by comparing the difference in cosine similarity between images and class-related description texts versus class-perturbed random texts. The detailed algorithm and pseudo-code of these three compared multi-modal detection methods are provided in Appendix D, while implementation details for our proposed Subspace Detection are provided in Appendix E.2.

**Evaluation metrics.**   Following existing evaluation metrics adopted by Gao et al. (2019) and Liu et al. (2023b), we use two widely adopted metrics to evaluate the effectiveness of the proposed method: 1) Area Under Receiver Operating Curve (*i.e.*, AUROC), which evaluates the detection's ability to distinguish between poisoned and benign images by measuring the trade-off between the true positive rate and false positive rate across different thresholds. 2) F1-score, which evaluates the balance between precision and recall. A large AUROC value and a large F1-score value indicate that the detection can accurately distinguish between poisoned and benign images.

## 5.2   MAIN RESULTS

**Evaluation on unimodal detection methods.**   Results are shown in Tab. 2. **SCALE-UP** detects poisoned images by analyzing the consistency of predictions through multiple rounds of pixel-wise amplification, identifying those with high consistency as poisoned. Among all evaluated attacks, SCALE-UP achieves its best detection performance against BadNets, a traditional backdoor attack designed for single-modal, with an AUROC value of 0.895 and an F1-score value of 0.870 on ImageNet-1K and an AUROC value of 0.870 and an F1-score value of 0.851 on ImageNet-R. However, SCALE-UP struggles to detect backdoor attacks designed for multi-modal models. In addition, we observe that SCALE-UP shows reduced performance on more challenging datasets, such as ImageNet-R and ImageNet-Sketch (*e.g.*, an average AUROC of 0.543 on ImageNet-R and an average AUROC of 0.427 on ImageNet-Sketch ), compared to its performance on the standard ImageNet-1K dataset (*e.g.*, an average AUROC of 0.577). **STRIP** detects poisoned images by perturbing inputs and measuring the prediction entropy, identifying those with low entropy as poisoned. While STRIP demonstrates great performance under Blended backdoor attack on ImageNet-R, achieving an AUROC of 0.721 and an F1-score of 0.734, its overall effectiveness remains limited, with an average AUROC of 0.456 on ImageNet-1K, 0.480 on ImageNet-R, and 0.466 on ImageNet-Sketch.

**Evaluation on multi-modal detection methods.**   Results are shown in Tab. 2. **Descriptive Rephrasing** detects images by measuring the absolute difference between their similarities to the original class text and its descriptive rephrasing. Descriptive Rephrasing significantly outperforms unimodal detection methods in overall three datasets, *i.e.*, an average AUROC of 0.686 on ImageNet-1K, 0.651 and 0.649 on ImageNet-R and ImageNet-Sketch, respectively. Although slightly improved results have been achieved, further performance improvements are still expected. **Font Styling** detects images by measuring the absolute difference between their similarities to the original class text and its bold styling text. Font Styling demonstrates great detection performance across some

Table 2: Comparison with the state-of-the-art defenses on three datasets on ResNet-50. Note that the best result is highlighted in **boldface**. (Descriptive Rephrasing $w.r.t.$ Description, Font Styling $w.r.t.$ Font, Language Translation $w.r.t.$ Language.)

| Datasets | Detection Attack ↓ | SCALE-UP Guo et al. (2023) | | STRIP Gao et al. (2019) | | Description | | Font | | Language | | Subspace Detection | |
|---|---|---|---|---|---|---|---|---|---|---|---|---|---|
| | | AUROC | F1-score | AUROC | F1-score | AUROC | F1-score | AUROC | F1-score | AUROC | F1-score | AUROC | F1-score |
| ImageNet-1K (Deng et al., 2009) | BadNets(Gu et al., 2019) | 0.895 | 0.870 | 0.556 | 0.668 | 0.811 | 0.753 | 0.584 | 0.667 | 0.565 | 0.668 | **0.962** | **0.920** |
| | Blended(Chen et al., 2017) | 0.506 | 0.667 | 0.375 | 0.667 | 0.558 | 0.667 | 0.531 | 0.668 | 0.500 | 0.667 | **0.982** | **0.969** |
| | SIG(Barni et al., 2019) | 0.495 | 0.667 | 0.203 | 0.667 | 0.680 | 0.675 | 0.456 | 0.667 | 0.489 | 0.667 | **0.692** | **0.788** |
| | WaNet(Nguyen & Tran, 2021) | 0.660 | 0.667 | 0.520 | 0.672 | 0.721 | 0.695 | 0.853 | 0.797 | 0.818 | 0.764 | **0.931** | **0.901** |
| | TrojVQA(Walmer et al., 2022) | 0.502 | 0.667 | 0.622 | 0.667 | 0.635 | 0.667 | 0.512 | 0.667 | 0.534 | 0.667 | **0.925** | **0.879** |
| | Carlini & Terzis(Carlini & Terzis, 2022) | 0.489 | 0.667 | 0.508 | 0.667 | 0.688 | 0.672 | 0.438 | 0.667 | 0.413 | 0.667 | **0.994** | **0.968** |
| | BadCLIP(Liang et al., 2024) | 0.490 | 0.667 | 0.407 | 0.667 | 0.709 | 0.671 | 0.822 | 0.749 | 0.801 | 0.731 | **0.966** | **0.963** |
| | Average | 0.577 | 0.696 | 0.456 | 0.668 | 0.686 | 0.686 | 0.600 | 0.697 | 0.589 | 0.690 | **0.922** | **0.913** |
| ImageNet-R (Hendrycks et al., 2021) | BadNets(Gu et al., 2019) | 0.877 | 0.851 | 0.265 | 0.667 | 0.706 | 0.694 | 0.488 | 0.667 | 0.436 | 0.667 | **0.918** | **0.864** |
| | Blended(Chen et al., 2017) | 0.468 | 0.667 | 0.721 | 0.734 | 0.533 | 0.686 | 0.540 | 0.671 | 0.589 | 0.681 | **0.884** | **0.919** |
| | SIG(Barni et al., 2019) | 0.420 | 0.667 | 0.418 | 0.667 | 0.597 | 0.670 | 0.234 | 0.667 | 0.322 | 0.667 | **0.625** | **0.769** |
| | WaNet(Nguyen & Tran, 2021) | 0.589 | 0.667 | 0.628 | 0.688 | 0.712 | 0.690 | 0.820 | 0.778 | 0.790 | 0.763 | **0.826** | **0.858** |
| | TrojVQA(Walmer et al., 2022) | 0.477 | 0.667 | 0.347 | 0.667 | 0.703 | 0.682 | 0.453 | 0.667 | 0.526 | 0.667 | **0.865** | **0.856** |
| | Carlini & Terzis(Carlini & Terzis, 2022) | 0.488 | 0.667 | 0.507 | 0.677 | 0.562 | 0.667 | 0.361 | 0.667 | 0.353 | 0.667 | **0.992** | **0.957** |
| | BadCLIP(Liang et al., 2024) | 0.481 | 0.667 | 0.473 | 0.667 | 0.744 | 0.680 | 0.778 | 0.720 | 0.747 | 0.693 | **0.896** | **0.903** |
| | Average | 0.543 | 0.693 | 0.480 | 0.681 | 0.651 | 0.681 | 0.525 | 0.691 | 0.538 | 0.686 | **0.858** | **0.875** |
| ImageNet-Sketch (Wang et al., 2019) | BadNets(Gu et al., 2019) | 0.526 | 0.667 | 0.356 | 0.667 | 0.711 | 0.695 | 0.479 | 0.667 | 0.444 | 0.667 | **0.922** | **0.887** |
| | Blended(Chen et al., 2017) | 0.420 | 0.667 | 0.568 | 0.698 | 0.548 | 0.678 | 0.533 | 0.668 | 0.577 | 0.674 | **0.890** | **0.916** |
| | SIG(Barni et al., 2019) | 0.347 | 0.667 | 0.379 | 0.673 | 0.582 | 0.667 | 0.247 | 0.667 | 0.339 | 0.667 | **0.621** | **0.769** |
| | WaNet(Nguyen & Tran, 2021) | 0.454 | 0.667 | 0.414 | 0.667 | 0.699 | 0.687 | 0.818 | 0.782 | 0.795 | 0.766 | **0.882** | **0.859** |
| | TrojVQA(Walmer et al., 2022) | 0.429 | 0.667 | 0.427 | 0.667 | 0.700 | 0.677 | 0.459 | 0.667 | 0.517 | 0.667 | **0.863** | **0.854** |
| | Carlini & Terzis(Carlini & Terzis, 2022) | 0.381 | 0.667 | 0.618 | 0.713 | 0.563 | 0.667 | 0.364 | 0.667 | 0.374 | 0.667 | **0.980** | **0.948** |
| | BadCLIP(Liang et al., 2024) | 0.430 | 0.667 | 0.502 | 0.670 | 0.740 | 0.681 | 0.777 | 0.720 | 0.753 | 0.703 | **0.954** | **0.939** |
| | Average | 0.427 | 0.667 | 0.466 | 0.679 | 0.649 | 0.679 | 0.525 | 0.691 | 0.543 | 0.687 | **0.873** | **0.882** |

attack methods. For example, on the ImageNet-1K dataset, it achieves an AUROC of 0.853 and an F1-score of 0.797 against the WaNet attack, and an AUROC of 0.822 with an F1-score of 0.749 against the BadCLIP attack. On ImageNet-R, Font Styling achieves an AUROC of 0.820 and an F1-score of 0.778 for WaNet, and an AUROC of 0.778 with an F1-score of 0.720 for BadCLIP. Similarly, on ImageNet-Sketch, it reaches an AUROC of 0.818 and an F1-score of 0.782 against WaNet, and an AUROC of 0.777 with an F1-score of 0.720 against BadCLIP. However, its performance is comparatively poor against other attacks, indicating that this textual transformation lacks generality across all attacks. **Language Translation** detects images by measuring the absolute difference between their similarities to the original class text and its Arabic translation. Similar to Font Styling, Language Translation demonstrates strong detection performance against WaNet and BadCLIP attacks. However, its effectiveness varies across different types of attacks, indicating these methods lack generality. **Subspace Detection** detects images by measuring the Euclidean distance between their visual features and a set of text variant features sampled from the subspace. Subspace Detection consistently demonstrates superior detection performance across all evaluation scenarios, encompassing a wide range of datasets and attacks. On the ImageNet-1K, it achieves an excellent average AUROC of 0.922 and an average F1-score of 0.913. While its performance exhibits slight variations on more challenging datasets such as ImageNet-R and ImageNet-Sketch, the method still outperforms other detection methods, achieving an average AUROC of 0.858 and F1-score of 0.875 on ImageNet-R, and an AUROC of 0.873 and F1-score of 0.882 on ImageNet-Sketch. These results validate the effectiveness and generality of Subspace Detection.

**Additional Evaluations and Analysis in Appendix.** Due to space limitations, several additional evaluations and analysis are provided in Appendix F, including: (i) results based on the CLIP model with a **ViT-B/32** visual encoder (Sec. F.1); (ii) performance against **adaptive attacks** (Sec. F.2); and (iii) analysis of **computational cost** (Sec. F.4). (iv) Comparisons with the BDetCLIP method are provided in the Appendix F.4.

## 5.3 Ablation Studies

**Impact of sampling along positive direction.** In step ③ of Sec. 4.2.1, we define the positive direction as the vector from original text feature to text variant feature. Conversely, the

Table 3: Impact of sampling along positive direction.

| Attack Direction ↓ | BadNetsGu et al. (2019) | | Carlini & TerzisCarlini & Terzis (2022) | | WaNetNguyen & Tran (2021) | |
|---|---|---|---|---|---|---|
| | AUROC | F1-score | AUROC | F1-score | AUROC | F1-score |
| Positive | 0.962 | 0.920 | 0.994 | 0.968 | 0.931 | 0.901 |
| Negative | 0.525 | 0.703 | 0.702 | 0.749 | 0.246 | 0.701 |

reverse direction is referred to as the negative direction. To effectively examine the impact of direction selection, we investigate the performance of our detection method under both positive direction and negative direction. As shown in Tab. 3, there is a significant difference in detection effectiveness between the two sampling directions, indicating that the choice of direction plays a crucial role in

Figure 4: Impact of region-of-interest modeling times.

sampling discriminative text variant features. As a result, the positive direction is adopted in our method to ensure optimal detection performance.

**Impact of region-of-interest modeling times $L$.** As mentioned in step ③ of Sec. 4.2.1, repeating the entire process, *i.e.*, sampling, filtering, and modeling, $L$ times yields $L$ Gaussian distributions, which induces a uniform mixture of Gaussians $p_{mix}$. To explore the impact of different yielding times $L$ on detection performance, we conduct experiments under diverse attacks with the number of times ranging from 1 to 4. As shown in Fig. 4, the detection performance exhibits a clear upward trend as the number of yielding times $L$ increases, especially when increasing the yielding times from one to two. Therefore, increasing the yielding times within a limited range can effectively improve the detection performance. Considering the computational cost, we set the yielding times to 3 (*i.e.*, $L = 3$) in our experiments.

**Impact of individual text transformation.** We examine manifold approximation under different text transformations, including single types, pairwise combinations, and all three jointly. As shown in Tab. 4, detection performance improves as number of transformation types increases. Notably, combining two transformations improves the minimum detection performance over any single transformation, while three transformations increase it even more. This indicates performance improvement mainly arises from the synergy among transformations rather than a single transformation alone.

Table 4: Impact of individual text transformation.

| Attack Transformation↓ | BadNets Gu et al. (2019) AUROC | F1-score | Blended Chen et al. (2017) AUROC | F1-score | WaNet Nguyen & Tran (2021) AUROC | F1-score | BadCLIP Liang et al. (2024) AUROC | F1-score |
|---|---|---|---|---|---|---|---|---|
| Language | 0.838 | 0.821 | 0.999 | 0.996 | 0.999 | 0.999 | 0.999 | 0.997 |
| Font | 0.924 | 0.878 | 0.973 | 0.939 | 0.875 | 0.846 | 0.954 | 0.924 |
| Description | 0.749 | 0.802 | 0.923 | 0.882 | 0.543 | 0.754 | 0.949 | 0.902 |
| Minimum(Single) | 0.749 | 0.802 | 0.923 | 0.882 | 0.543 | 0.754 | 0.949 | 0.902 |
| Font+Description | 0.953 | 0.917 | 0.973 | 0.959 | 0.913 | 0.887 | 0.968 | 0.961 |
| Language+Font | 0.944 | 0.889 | 0.980 | 0.960 | 0.911 | 0.871 | 0.957 | 0.951 |
| Description+Language | 0.768 | 0.805 | 0.923 | 0.895 | 0.550 | 0.760 | 0.956 | 0.910 |
| Minimum(Pairwise) | 0.768 | 0.805 | 0.923 | 0.895 | 0.550 | 0.760 | 0.956 | 0.910 |
| All | 0.962 | 0.920 | 0.982 | 0.969 | 0.931 | 0.901 | 0.966 | 0.963 |

**Impact of the number of augmented text features for modeling $n$.** As reported in Tab. 5, increasing $n$ consistently improves both AUROC and F1-score for detection. A larger $n$ provides more augmented text features during manifold approximation, which helps approximate a more diverse semantic distribution. Consequently, when detection is performed, it becomes more favorable to sample diverse semantic text features from the distribution, enabling more effective discrimination between benign and poisoned images.

**Impact of the number of sampled text features for detection $n_s$.** As shown in Tab. 6, increasing $n_s$ steadily improves both AUROC and F1-score for detection. The results indicate that sampling more text features to distinguish between benign and poisoned images helps capture features that are semantically similar to the target text but not aligned with poisoned images. This reduces the risk of misjudgment and ensures the effectiveness of the detection.

Table 5: Impact of the number of augmented text features for modeling $n$.

| Attack $n$ ↓ | BadNets Gu et al. (2019) AUROC | F1-score | WaNet Nguyen & Tran (2021) AUROC | F1-score |
|---|---|---|---|---|
| 18 | 0.925 | 0.896 | 0.912 | 0.889 |
| 54 | 0.962 | 0.920 | 0.931 | 0.901 |
| 90 | 0.972 | 0.925 | 0.939 | 0.913 |

Table 6: Impact of the number of sampled text features for detection $n_s$.

| Attack $n_s$ ↓ | BadNets Gu et al. (2019) AUROC | F1-score | WaNet Nguyen & Tran (2021) AUROC | F1-score |
|---|---|---|---|---|
| 9 | 0.925 | 0.876 | 0.912 | 0.871 |
| 15 | 0.962 | 0.920 | 0.931 | 0.901 |
| 21 | 0.972 | 0.925 | 0.939 | 0.913 |

**Impact of the manifold approximation.** To assess the impact of manifold approximation, we employ single text transformations and use the Euclidean distance (ED) as the detection metric. As shown

Table 7: Impact of the manifold approximation.

| Attack Detection↓ | Blended Chen et al. (2017) | | WaNet Nguyen & Tran (2021) | | BadCLIP Liang et al. (2024) | |
|---|---|---|---|---|---|---|
| | AUROC | F1-score | AUROC | F1-score | AUROC | F1-score |
| Description \ w ED | 0.776 | 0.845 | 0.421 | 0.739 | 0.593 | 0.754 |
| Font \ w ED | 0.811 | 0.785 | 0.561 | 0.712 | 0.539 | 0.685 |
| Language \ w ED | 0.740 | 0.799 | 0.260 | 0.688 | 0.493 | 0.710 |
| Subspace Detection | 0.982 | 0.969 | 0.931 | 0.901 | 0.966 | 0.963 |

in Tab. 7, Subspace Detection substantially outperforms the individual transformations, indicating the necessity of manifold approximation.

# 6 CONCLUSION

In this work, we revealed the phenomenon of *shallow malicious matching* for poisoned images, in contrast to *deep benign matching* for benign images in backdoored CLIP. Building on this finding, we propose a test-time poisoned image detection method against backdoored CLIP, named **Subspace Detection**. By building a subspace lay on the manifold and biased toward a region-of-interest densely related to detection effectiveness, we distinguish poisoned from benign images based on their Euclidean distances with text features from this region. Experimental results demonstrate that the proposed method achieves strong effectiveness and generality across diverse attacks and datasets, outperforming prior detection methods.

# 7 ETHICS STATEMENT

In this paper, we identify a potential security risk in CLIP models stemming from backdoor attacks, which can cause misclassification of poisoned images containing trigger patterns. Ensuring the security of CLIP is important due to its popularity. The proposed Subspace Detection aims to detect and reject poisoned inputs at test time. Evaluation results demonstrate that our method can enhance the security of CLIP-based systems.

While we strive to improve CLIP's robustness against backdoor attacks, we acknowledge that Subspace Detection does not provide complete immunity to all threats. Moreover, like any detection method, there exists a risk of misuse by attackers who may exploit its weaknesses and attempt to bypass the detection process. By sharing our findings and methodology, we hope to encourage the ongoing development of more robust and reliable security frameworks for CLIP.

# 8 REPRODUCIBILITY STATEMENT

The complete source code for our proposed Subspace Detection is provided in the supplementary materials. Our experimental setup, including the specific CLIP models, downstream datasets, and the configurations of all evaluated backdoor attacks, is described in Sec. 5.1. Furthermore, detailed implementation settings and key hyperparameters for our method are provided in Appendix E.2. We believe these resources offer a clear and complete way to reproduce our results and facilitate future research in this area.

ACKNOWLEDGMENTS

Baoyuan Wu is supported by Guangdong Basic and Applied Basic Research Foundation (No. 2024B1515020095), Guangdong Provincial Program (No. 2023TQ07A352), Shenzhen Science and Technology Program (No. RCYX20210609103057050 and JCYJ20240813113608011), and Longgang District Key Laboratory of Intelligent Digital Economy Security.

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

## ORGANIZATION OF THE APPENDIX

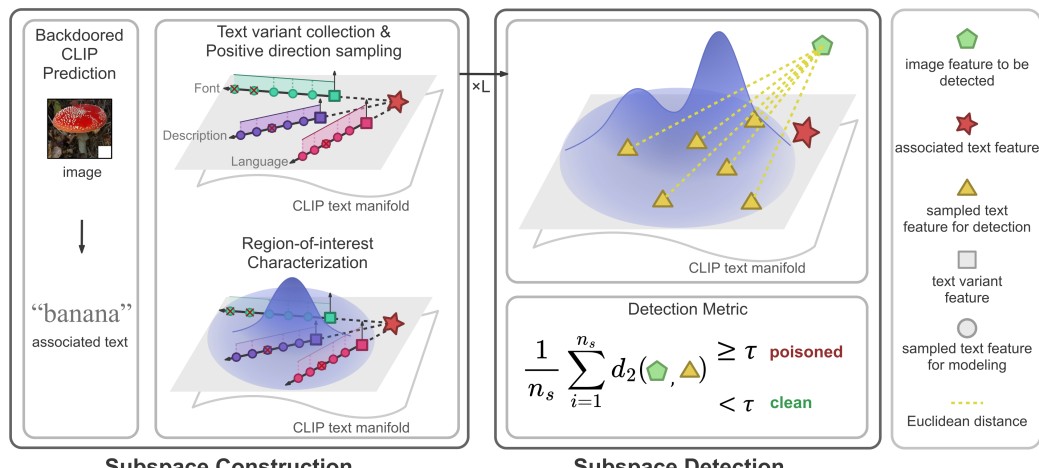

Figure 5: The workflow of test-time poisoned image detection via *Subspace Detection*.

The workflow of the proposed *Subspace Detection* is illustrated in Fig. 5. We have also provided the table of contents below to better navigate the content in the appendix.

Sec. A provides a comprehensive overview of related work for defense methods against backdoor attacks for CLIP models.

Sec. B provides the implementation details of the detection threshold determination in the proposed Subspace Detection.

Sec. C provides the pseudo-code of the proposed Subspace Detection.

Sec. D provides the detailed algorithm and the pseudo-code of three compared multi-modal detection methods (*i.e.*, Descriptive Rephrasing, Font Styling and Language Translation) in Sec. 5 of the main text.

Sec. E provides implementation details of the backdoor attacks and detections.

Sec. F provides more experimental results, such as the results based on the open-source CLIP model using the ViT-B/32 visual encoder, the results against the adaptive attack, the discussion on the motivation for choosing the Euclidean distance, the analysis about time cost.

Sec. G provides the limitation of the proposed method.

Sec. H provides the role of the LLM in this work.

## A    RELATED WORK OF BACKDOOR DEFENSES ON CLIP

**Defense at pre-training time.** The defender at pre-training time aims to train a CLIP model with high benign accuracy while simultaneously mitigating backdoor injection based on a suspicious pre-training dataset. RoCLIP (Yang et al., 2023) periodically matches each image in the pre-training dataset to its nearest-neighbor caption from a dynamic pool—rather than its original paired caption—and applies data augmentation to further disrupt malicious correlations. SafeCLIP (Yang et al., 2024) also proposes a robust pre-training framework by using a unimodal warmup phase to separate pre-training dataset into safe and risky subsets, which are then handled with a mixed training strategy of cross-modal and unimodal losses, to break backdoor attacks.

**Defense at post-training time.** Given an untrustworthy model that may contain backdoors, the defender at post-training time aims to obtain a secure model by removing the potential backdoor while preserving the benign performance (Bansal et al., 2023; Xun et al., 2024; Kuang et al., 2024; Zhang et al., 2024). CleanCLIP (Bansal et al., 2023) finetunes the backdoored model by incorporating

a unimodal self-supervised objective to learn robust features for each modality independently, thus breaking the spurious correlations between triggers and target labels learned during pre-training.

**Defense at test time.** To prevent a backdoor from being activated by querying with the specific trigger pattern at test time, defense strategies can be achieved by *robust prediction* and *poisoned sample detection*. Robust prediction methods aim to give correct predictions even on poisoned samples, typically through poisoned sample relabeling (Huang et al., 2023; Liu et al., 2023a) or purification (Shi et al., 2023). For poisoned sample detection, please refer to Sec. 2.3.

## B    IMPLEMENTATION DETAILS OF THE THRESHOLD

Following previous work  Liu et al. (2023b); Niu et al. (2024), the defender can empirically determine the threshold value $\tau'$ based on access to a small benign reference set $\mathcal{D}_{\text{ref}}$:

$$\tau' = \max_{\boldsymbol{v}_{\text{ref}} \in \mathcal{D}_{\text{ref}}} \left( \frac{1}{n_s} \sum_{i=1}^{n_s} d_2(\hat{f}_v(\boldsymbol{v}_{\text{ref}}) - \boldsymbol{z}_d^{\text{ref}(i)}) \right), \tag{2}$$

where the generation process of $\boldsymbol{z}_d^{\text{ref}}$ based on $\boldsymbol{v}_{\text{ref}}$ is similar to that of $\boldsymbol{z}_d$ based on $\boldsymbol{v}$, *i.e.*, sampling from the region-of-interest on $\boldsymbol{v}_{\text{ref}}$'s corresponding local text manifold. We select the largest distance from the local text manifold of a reference image $\boldsymbol{v}_{\text{ref}}$ over $\mathcal{D}_{\text{ref}}$ as the threshold.

## C    DETAILED ALGORITHM OF THE PROPOSED SUBSPACE DETECTION

The pseudo-code of the proposed Subspace Detection is summarized in Algorithm 1.

---

**Algorithm 1** Workflow of the proposed *Subspace Detection* for test-time poisoned image detection.

**Require:** Backdoored CLIP with text encoder $\hat{f}_t(\cdot; \hat{\boldsymbol{\theta}}_t)$ and image encoder $\hat{f}_v(\cdot; \hat{\boldsymbol{\theta}}_v)$, test dataset $\mathcal{D}_{\text{test}}$ and corresponding label names $\{\boldsymbol{t}^{(i)}\}_{i=1}^C$, $m$ text transformation functions including font transformations $\{T_f^{(i)}(\cdot)\}_{i=1}^{m_f}$, language transformations $\{T_l^{(i)}(\cdot)\}_{i=1}^{m_l}$ and description transformations $\{T_d^{(i)}(\cdot)\}_{i=1}^{m_d}$, subspace dimension $K$, the number of subspaces $L$, scale constant $\eta$, the number of augmented text features for modeling $n$, the number of sampled text features for detection $n_s$, threshold $\tau'$.

**for** test image $\boldsymbol{v} \in \mathcal{D}_{\text{test}}$ **do**

    Predict the label name for $\boldsymbol{v}$ on backdoored CLIP by $\boldsymbol{t} = \arg\max_{\boldsymbol{t}^{(c)}} p(\boldsymbol{t} = \boldsymbol{t}^{(c)}|\boldsymbol{v}) = \frac{\exp(\cos(\hat{f}_v(\boldsymbol{v}), \hat{f}_t(\boldsymbol{t}^{(c)}))/\tau)}{\sum_{j=1}^C \exp(\cos(\hat{f}_v(\boldsymbol{v}), \hat{f}_t(\boldsymbol{t}^{(j)}))/\tau)}$;

    Compute the feature of predicted text $\boldsymbol{z}_t = \hat{f}_t(\boldsymbol{t})$;
    Initialize the set $\mathcal{Z}_t \leftarrow \{\boldsymbol{z}_t\}$, the probability $p_{mix} \leftarrow 0$;
    **for** text transformation $T^{(i)}(\cdot) \in \{T_f^{(i)}(\cdot)\}_{i=1}^{m_f} \cup \{T_l^{(i)}(\cdot)\}_{i=1}^{m_l} \cup \{T_d^{(i)}(\cdot)\}_{i=1}^{m_d}$ **do**          ▷ [Variants collection]
        Generate a variant of $\boldsymbol{t}$ by $T^{(i)}(\cdot)$ and compute its feature $\boldsymbol{z}_t'^{(i)} = \hat{f}_t(T^{(i)}(\boldsymbol{t}))$;
        Append $\boldsymbol{z}_t'^{(i)}$ to $\mathcal{Z}_t$;
    **end for**
    Compute the mean $\boldsymbol{\mu}_t$ and top $K$ principal components $U_K$ of $\mathcal{Z}_t$ using PCA;          ▷ [Manifold approximation]
    Project $\boldsymbol{z}_{t,K} = \Pi_K(\boldsymbol{z}_t)$ and $\mathcal{Z}_{t,K}' = \{\Pi_K(\boldsymbol{z}_t'^{(i)})\}_{i=1}^m$ with $\Pi_K(\mathbf{u}) = U_K^\top(\mathbf{u} - \boldsymbol{\mu}_t)$;
    **for** $l = 1, 2, \ldots, L$ **do**          ▷ [Region-of-interest characterization]
        Initialize the set $\mathcal{Z}_{t,K}'' \leftarrow \mathcal{Z}_{t,K}'$;
        **for** $\boldsymbol{z}_{t,K}' \in \mathcal{Z}_{t,K}'$ **do**          ▷ [Positive direction sampling]
            **for** $t = 1, 2, \ldots, \frac{n}{m}$ **do**
                Sample along the positive direction $\boldsymbol{z}_{t,K}'' \leftarrow \boldsymbol{z}_{t,K}' + \alpha(\boldsymbol{z}_{t,K}' - \boldsymbol{z}_{t,K})$, $\alpha \sim U(0,1)$, subject to the constraint $0.85 < \frac{\cos(\Pi_d(\boldsymbol{z}_{t,K}''), \boldsymbol{z}_t)}{\cos(\Pi_d(\boldsymbol{z}_{t,K}'), \boldsymbol{z}_t)} < 1.0$;
                Append $\boldsymbol{z}_{t,K}''$ to $\mathcal{Z}_{t,K}''$;
            **end for**
        **end for**
        Project elements in $\mathcal{Z}_{t,K}''$ with $\Pi_d(\mathbf{u}) = \boldsymbol{\mu}_t + U_K \Pi_K(\mathbf{u})$ and obtain $\mathcal{Z}_{t,\mathcal{S}}'' = \{\boldsymbol{z}_{t,\mathcal{S}}''^{(i)}\}_{i=1}^{m+n}$;
        Compute the mean $\boldsymbol{\mu}_{\text{bias}}^{(l)}$ of $\mathcal{Z}_{t,\mathcal{S}}''$ and the deviation matrix $\boldsymbol{D}_{\text{bias}}^{(l)} = [\boldsymbol{z}_{t,\mathcal{S}}''^{(1)} - \boldsymbol{\mu}_{\text{bias}}, \ldots, \boldsymbol{z}_{t,\mathcal{S}}''^{(m+n)} - \boldsymbol{\mu}_{\text{bias}}]$;
        Update $p_{mix} \leftarrow \frac{l \times p_{mix} + \mathcal{N}(\boldsymbol{\mu}_{\text{bias}}^{(l)}, \eta^2 \boldsymbol{D}_{\text{bias}}^{(l)} \boldsymbol{D}_{\text{bias}}^{(l)\top})}{l+1}$;
    **end for**
    Sample $n_s$ text features $\{\boldsymbol{z}_d^{(i)}\}_{i=1}^{n_s}$ from $p$ and compute $\bar{\text{dis}} \leftarrow \frac{1}{n_s} \sum_{i=1}^{n_s} d_2(\boldsymbol{z}_v - \boldsymbol{z}_d^{(i)})$;          ▷ [Subspace detection]
    **if** $\bar{\text{dis}} \geq \tau'$ **then**
        **return** $\boldsymbol{v}$'s detection result "poisoned";
    **else**
        **return** $\boldsymbol{v}$'s detection result "benign".
    **end if**
**end for**

---

# D  DETAILED ALGORITHM OF MULTI-MODAL DETECTION METHODS

In this section, we provide the detailed algorithms of three compared multi-modal detection methods (*i.e.*, *Descriptive Rephrasing*, *Font Styling* and *Language Translation*) in Sec. 5 of the main text, which use the change in cosine similarity before and after text transformation as the detection metric. In particular, for an image $v$, we apply a text transformation $T(\cdot)$ on its associated text $t$ predicted by a backdoored CLIP, such as descriptive rephrasing $T_d(\cdot)$, font styling $T_f(\cdot)$ or language translation $T_l(\cdot)$, leading to transformed text $t' = T(t)$. The transformed text $t'$ is a variant of the original text $t$ that shares the same semantic concept, while its correlation with $v$ is not explicitly learned during the backdoor learning. We then calculate the cosine similarity between the image feature $z_v = \hat{f}_v(v)$ and the associated text feature $z_t = \hat{f}_t(t)$, *i.e.*, $S = \cos(z_v, z_t)$, and the cosine similarity between $z_v$ and the transformed text feature $z'_t = \hat{f}_t(T(t))$, *i.e.*, $S' = \cos(z_v, z'_t)$. Since the shallowness will hinder malicious matching from generalizing to semantically equivalent variants of the target text, while the deepness will help benign matching remain robust to text modifications, a promising detection strategy comparable to our Subspace Detection is to consider the change in cosine similarity before and after text transformation. According to the employed text transformation, the corresponding methods are called *Descriptive Rephrasing*, *Font Styling* and *Language Translation*. For any to-be-detected image $v$, if it satisfies the following condition:

$$| \cos\left(z_v, z_t\right) - \cos\left(z_v, z'_t\right) | \ge \epsilon, \tag{3}$$

it will be identified as poisoned; otherwise, it will be considered benign. $\epsilon$ is a threshold which could also be empirically determined based on $\mathcal{D}_{\text{ref}}$:

$$\epsilon = \max_{v_{\text{ref}} \in \mathcal{D}_{\text{ref}}} \left(| \cos\left(\hat{f}_v\left(v_{\text{ref}}\right), \hat{f}_t\left(t_{\text{ref}}\right)\right) - \cos\left(\hat{f}_v\left(v_{\text{ref}}\right), \hat{f}_t\left(T\left(t_{\text{ref}}\right)\right)\right) |\right), \tag{4}$$

where $t_{\text{ref}}$ is the associated text of $v_{\text{ref}}$ predicted by a backdoored CLIP. The pseudo-code is summarized in Algorithm 2.

---

**Algorithm 2** Workflow of the compared multi-modal detection methods.

---

**Require:** Backdoored CLIP with text encoder $\hat{f}_t(\cdot; \hat{\theta}_t)$ and image encoder $\hat{f}_v(\cdot; \hat{\theta}_v)$, test dataset $\mathcal{D}_{\text{test}}$ and corresponding label names $\{t^{(i)}\}_{i=1}^C$, text transformation function $T(\cdot)$, threshold $\epsilon$.
**for** test image $v \in \mathcal{D}_{\text{test}}$ **do**
    Predict the label name for $v$ on backdoored CLIP by $t = \arg\max_{t^{(c)}} p(t = t^{(c)}|v) = \frac{\exp(\cos(\hat{f}_v(v), \hat{f}_t(t^{(c)}))/\tau)}{\sum_{j=1}^C \exp(\cos(\hat{f}_v(v), \hat{f}_t(t^{(j)}))/\tau)}$;
    Compute the feature of predicted text $z_t = \hat{f}_t(t)$;
    Generate a variant of $t$ by $T(\cdot)$ and compute its feature $z'_t = \hat{f}_t(T(t))$;
    Compute $\Delta S \leftarrow | \cos\left(z_v, z_t\right) - \cos\left(z_v, z'_t\right) |$;
    **if** $\Delta S \ge \epsilon$ **then**
        **return** $v$'s detection result "poisoned";
    **else**
        **return** $v$'s detection result "benign".
    **end if**
**end for**

---

# E  IMPLEMENTATION DETAILS OF THE BACKDOOR ATTACKS AND DETECTIONS

## E.1  BACKDOOR ATTACKS SETTINGS

In our implementation of the backdoor attacks, we follow the training procedure and hyperparameter settings used in BadCLIP (Liang et al., 2024), except for the number of poisoned samples and the method used to generate poisoned samples in the compared backdoor attacks. Specifically, we use 500K image-text pairs from CC3M, including 5,000 poisoned samples, in the data poisoning phase. The details of the poisoned sample generation methods for the compared backdoor attacks are as follows:

- **BadNets:** Following Gu et al. (2019), we used the white square as the trigger. Specifically, a white square trigger of size 32×32 pixels was placed at the bottom-right corner of each original image to generate poisoned samples. A white square trigger was placed at the bottom-right corner of each original image to generate poisoned samples.
- **Blended:** Following Chen et al. (2017), we used the "HelloKitty" image as the trigger, then generate poisoned images by blending the original image and the trigger image with weights of 0.6 and 0.4, respectively.
- **SIG:** Following Barni et al. (2019), we opted for an horizontal sinusoidal signal as the trigger, and the noise amplitude and the frequency were set to 0.235 and $2\pi j \times 6/224$, respectively.
- **WaNet:** Following Nguyen & Tran (2021), the poisoned images were generated by applying a fixed elastic warping field to benign images, which subtly alters the spatial structure while preserving visual similarity. The control-grid size and the warping strength were set to 224 and 1, respectively.
- **TrojVQA:** Following Walmer et al. (2022), triggers were embedded separately in both the visual and textual modalities, implementing a dual-modal backdoor attack on the CLIP model by combining a centrally placed visual patch with a specific textual prompt ("remember") to jointly activate the backdoor.
- **Carlini & Terzis:** Following Carlini & Terzis (2022), poisoned images were generated by replacing a randomly chosen region in benign images with a 16×16 patch. The captions for poisoned images were randomly selected from a set of relevant descriptions corresponding to the target class within the training dataset.
- **BadCLIP:** Following Liang et al. (2024), the trigger generation process was optimized using natural language descriptions, with keeping parameter settings consistent with the original work.

### E.2 BACKDOOR DETECTIONS SETTINGS

- **SCALE-UP:** Following Guo et al. (2023), we amplified the pixel values of input images by five different factors, $i.e.$, 3, 5, 7, 9, 11. The key idea lies in this method is that the consistency of the model's predictions across these scales was used to identify whether the input image is poisoned.
- **STRIP:** Following Gao et al. (2019), we selected 64 benign samples to overlay on the input images and computed their prediction entropy, which was used as a threshold to determine whether other input images were poisoned.
- **Descriptive Rephrasing:** Following Algorithm 2, we used GPT-4 to generate descriptions based on the characteristics of each class without explicitly including the class name. (The prompt example is shown in the Fig. 6.) The defender can empirically determine the threshold value $\epsilon$ based on access to a small benign reference set $\mathcal{D}_{\text{ref}}$.

---

**Prompt: Generate 6 descriptions of "banana" that highlight its distinctive characteristics without using the word "banana". The description should be specific enough for someone to infer the object being described.**

**Answer:**
- "A fruit with a peel that is typically yellow when ripe."
- "A sweet fruit with a soft, creamy texture."
- "A nutritious fruit that can be eaten raw or used in smoothies."
- "A fruit known for its easy-to-peel skin and distinctive shape."
- "A tropical fruit that is high in potassium."
- "A yellow, curved fruit commonly eaten as a snack."

---

Figure 6: Prompt for generating descriptive rephrasings of "banana" using GPT-4.

- **Font Styling:** In Algorithm 2, we replaced the font style of the letters in the class name, such as using bold or italic fonts. (The font examples are shown in the Fig. 7.) In the compared experiment, we adopted bold font. Consistent with the Descriptive Rephrasing, the defender can empirically determine the threshold value $\epsilon$ based on access to a small benign reference set $\mathcal{D}_{\text{ref}}$.
- **Language Translation:** In Algorithm 2, we used GPT-4 to translate class name into Arabic. For example, given the class name "banana", the prompt provided to GPT-4 was: *"Translate the word*

| a→**a** | b→**b** | c→**c** | d→**d** | e→**e** | f→**f** | g→**g** | h→**h** | i→**i** | j→**j** | k→**k** | l→**l** | m→**m** |
|---|---|---|---|---|---|---|---|---|---|---|---|---|
| n→**n** | o→**o** | p→**p** | q→**q** | r→**r** | s→**s** | t→**t** | u→**u** | v→**v** | w→**w** | x→**x** | y→**y** | z→**z** |
| A→**A** | B→**B** | C→**C** | D→**D** | E→**E** | F→**F** | G→**G** | H→**H** | I→**I** | J→**J** | K→**K** | L→**L** | M→**M** |
| N→**N** | O→**O** | P→**P** | Q→**Q** | R→**R** | S→**S** | T→**T** | U→**U** | V→**V** | W→**W** | X→**X** | Y→**Y** | Z→**Z** |

(a) Bold Font

| a→*a* | b→*b* | c→*c* | d→*d* | e→*e* | f→*f* | g→*g* | h→*h* | i→*i* | j→*j* | k→*k* | l→*ℓ* | m→*m* |
|---|---|---|---|---|---|---|---|---|---|---|---|---|
| n→*n* | o→*o* | p→*p* | q→*q* | r→*r* | s→*s* | t→*t* | u→*u* | v→*v* | w→*w* | x→*x* | y→*y* | z→*z* |
| A→*𝒜* | B→*ℬ* | C→*𝒞* | D→*𝒟* | E→*ℰ* | F→*ℱ* | G→*𝒢* | H→*ℋ* | I→*𝒥* | J→*𝒥* | K→*𝒦* | L→*ℒ* | M→*𝑀* |
| N→*𝒩* | O→*𝒪* | P→*𝒫* | Q→*𝒬* | R→*ℛ* | S→*𝒮* | T→*𝒯* | U→*𝒰* | V→*𝒱* | W→*𝒲* | X→*𝒳* | Y→*𝒴* | Z→*𝒵* |

(b) Italic Font

Figure 7: Letters with corresponding font style transformations: (a) Bold font transformation; (b) Italic font transformation.

*'banana' into Arabic.*" Consistent with the Descriptive Rephrasing, the defender can empirically determine the threshold value $\epsilon$ based on access to a small benign reference set $\mathcal{D}_{\text{ref}}$.

- **BDetCLIP:** Following Niu et al. (2024), we used 7 class-related description texts and 1 class-perturbed random text for each class. Clean and poisoned images were distinguished by computing the difference between the average cosine similarity of an image with 7 class-related description texts and the cosine similarity of the image with 1 class-perturbed random text. The defender can empirically determine the threshold value based on access to a small benign reference set $\mathcal{D}_{\text{ref}}$.
- **Subspace Detection:** In Algorithm 1, The numbers of font, language, and description transformations are set to $m_f = 2$, $m_l = 1$, and $m_d = 6$, respectively, leading to a total number of text transformations $m = 9$. The dimension of the subspace $K$ is selected to retain 95% of the total variance. The number of subspaces $L = 3$. The number of augmented text features for modeling and the number of sampled text features for detection are set to $n = 90$ and $n_s = 15$. The scale constant $\eta$ in Gaussian distribution is set to 0.001. Consistent with the Descriptive Rephrasing, the defender can empirically determine the threshold value $\tau'$ based on access to a small benign reference set $\mathcal{D}_{\text{ref}}$.

# F  MORE EXPERIMENTAL RESULTS

## F.1  EXPERIMENTAL RESULTS UNDER THE VIT-B/32 VISUAL ENCODER.

To evaluate the effectiveness of the proposed Subspace Detection method, we also conduct experiments based on the open-source CLIP model (Ilharco et al., 2021), using ViT-B/32 (Dosovitskiy et al., 2020) as the visual encoder.

**Evaluation on unimodal detection methods.**  Results are shown in Tab. 8. **SCALE-UP** demonstrates excellent detection performance on single-modal models, but it shows relatively weaker performance on multi-modal models. Its ability to distinguish between poisoned and benign samples is limited according to the AUROC metric. Specifically, on the ImageNet-1K dataset, SCALE-UP achieves its highest AUROC of 0.615 against SIG attack. On both the ImageNet-R and ImageNet-Sketch datasets, the best performance is achieved against WaNet, with AUROC values of 0.552 and 0.511 respectively. In addition, we observe that SCALE-UP shows reduced performance on more challenging datasets, with average AUROC values of 0.517 on ImageNet-1K, 0.495 on ImageNet-R, and 0.452 on ImageNet-Sketch across all attacks. **STRIP** achieves average AUROC values of 0.497, 0.504, and 0.506 on ImageNet-1K, ImageNet-R, and ImageNet-Sketch, respectively, across all attacks. Although the detection performance is consistent across datasets, the overall effectiveness remains suboptimal.

**Evaluation on multi-modal detection methods.**  Results are shown in Tab. 8. **Descriptive Rephrasing** significantly outperforms unimodal detection methods in overall three datasets, *i.e.*, an average AUROC of 0.876 of ImageNet-1K, 0.789 and 0.826 on ImageNet-R and ImageNet-Sketch, respectively. Although there are slight variations in performance across different datasets, the overall performance remains strong. **Font Styling** achieves its highest AUROC of 0.555 against Blended

Table 8: Comparison with the state-of-the-art defenses on three datasets on ViT-B/32 (%). Note that the best result is highlighted in **boldface**. (Descriptive Rephrasing $w.r.t.$ Description, Font Styling $w.r.t.$ Font, Language Translation $w.r.t.$ Language.)

| Datasets | Detection | SCALE-UP | | STRIP | | Description | | Font | | Language | | Subspace Detection | |
|---|---|---|---|---|---|---|---|---|---|---|---|---|---|
| | Trigger | AUROC | F1-score | AUROC | F1-score | AUROC | F1-score | AUROC | F1-score | AUROC | F1-score | AUROC | F1-score |
| ImageNet-1K | BadNets | 0.477 | 0.667 | 0.485 | 0.667 | **0.847** | **0.787** | 0.552 | 0.693 | 0.381 | 0.667 | 0.750 | 0.722 |
| | Blended | 0.481 | 0.667 | 0.500 | 0.667 | 0.954 | 0.883 | 0.555 | 0.667 | 0.492 | 0.667 | **0.994** | **0.964** |
| | SIG | 0.615 | 0.667 | 0.525 | 0.667 | 0.973 | 0.914 | 0.541 | 0.667 | 0.495 | 0.667 | **0.986** | **0.947** |
| | WaNet | 0.520 | 0.667 | 0.500 | 0.667 | 0.934 | 0.874 | 0.440 | 0.667 | 0.374 | 0.667 | **0.952** | **0.882** |
| | TrojVQA | 0.494 | 0.667 | 0.476 | 0.667 | 0.818 | 0.776 | 0.370 | 0.667 | 0.253 | 0.667 | **0.925** | **0.850** |
| | Carlini & Terzis | 0.551 | 0.667 | 0.493 | 0.667 | 0.799 | 0.770 | 0.358 | 0.667 | 0.343 | 0.667 | **0.999** | **0.994** |
| | BadCLIP | 0.482 | 0.667 | 0.500 | 0.667 | 0.807 | 0.775 | 0.405 | 0.667 | 0.294 | 0.667 | **0.930** | **0.859** |
| | Average | 0.517 | 0.667 | 0.497 | 0.667 | 0.876 | 0.826 | 0.460 | 0.671 | 0.376 | 0.667 | **0.934** | **0.888** |
| ImageNet-R | BadNets | 0.471 | 0.667 | 0.498 | 0.667 | 0.742 | 0.734 | 0.323 | 0.667 | 0.307 | 0.667 | **0.771** | **0.744** |
| | Blended | 0.488 | 0.667 | 0.500 | 0.667 | 0.944 | 0.878 | 0.546 | 0.667 | 0.538 | 0.667 | **0.995** | **0.969** |
| | SIG | 0.509 | 0.667 | 0.530 | 0.667 | **0.970** | **0.908** | 0.555 | 0.667 | 0.609 | 0.668 | 0.959 | 0.898 |
| | WaNet | 0.552 | 0.667 | 0.520 | 0.667 | 0.870 | 0.815 | 0.420 | 0.667 | 0.402 | 0.667 | **0.947** | **0.880** |
| | TrojVQA | 0.469 | 0.667 | 0.483 | 0.667 | 0.676 | 0.713 | 0.321 | 0.667 | 0.260 | 0.667 | **0.933** | **0.863** |
| | Carlini & Terzis | 0.505 | 0.667 | 0.500 | 0.667 | 0.695 | 0.734 | 0.293 | 0.667 | 0.264 | 0.667 | **0.998** | **0.995** |
| | BadCLIP | 0.468 | 0.667 | 0.500 | 0.667 | 0.625 | 0.695 | 0.304 | 0.667 | 0.265 | 0.667 | **0.927** | **0.853** |
| | Average | 0.495 | 0.667 | 0.504 | 0.667 | 0.789 | 0.782 | 0.395 | 0.667 | 0.378 | 0.667 | **0.933** | **0.886** |
| ImageNet-Sketch | BadNets | 0.419 | 0.667 | 0.505 | 0.669 | 0.789 | 0.774 | 0.294 | 0.667 | 0.264 | 0.667 | **0.795** | **0.755** |
| | Blended | 0.501 | 0.667 | 0.510 | 0.667 | 0.958 | 0.887 | 0.484 | 0.667 | 0.420 | 0.667 | **0.990** | **0.955** |
| | SIG | 0.419 | 0.667 | 0.500 | 0.667 | **0.996** | **0.967** | 0.543 | 0.705 | 0.602 | 0.717 | 0.940 | 0.880 |
| | WaNet | 0.511 | 0.667 | 0.500 | 0.667 | 0.909 | 0.861 | 0.375 | 0.667 | 0.277 | 0.667 | **0.982** | **0.929** |
| | TrojVQA | 0.427 | 0.667 | 0.540 | 0.667 | 0.689 | 0.724 | 0.344 | 0.667 | 0.248 | 0.667 | **0.912** | **0.838** |
| | Carlini & Terzis | 0.486 | 0.667 | 0.490 | 0.667 | 0.761 | 0.755 | 0.289 | 0.667 | 0.204 | 0.667 | **0.998** | **0.993** |
| | BadCLIP | 0.402 | 0.667 | 0.500 | 0.667 | 0.678 | 0.713 | 0.335 | 0.667 | 0.253 | 0.667 | **0.882** | **0.817** |
| | Average | 0.452 | 0.667 | 0.506 | 0.667 | 0.826 | 0.812 | 0.381 | 0.672 | 0.324 | 0.674 | **0.928** | **0.881** |

attack on ImageNet-1K, and against SIG attack on ImageNet-R and ImageNet-Sketch, with AUROC values of 0.555 and 0.543, respectively. The method shows limited effectiveness under the ViT-B/32 model. **Language Translator** achieves its highest AUROC of 0.495 against SIG attack on ImageNet-1K, and performs best against SIG on both ImageNet-R and ImageNet-Sketch, with AUROC values of 0.609 and 0.602, respectively. However, the average AUROC across all attacks on ImageNet-1K, ImageNet-R, and ImageNet-Sketch is only 0.376, 0.378, and 0.324, respectively. This large performance gap across different attacks within the same dataset indicates that the method lacks generality. **Subspace Detection** demonstrates effective detection performance across all three datasets, achieving an average AUROC of 0.934 on ImageNet-1K and 0.933 on ImageNet-R. Moreover, even when evaluated on the more challenging ImageNet-Sketch dataset, it maintains a strong average AUROC of 0.928 across all attacks.

**Summary.** The proposed Subspace Detection method still demonstrates excellent detection performance when evaluated on the ViT-B/32 model, indicating that the method is not limited to a specific architecture and further supporting its practical generality in diverse settings.

## F.2 ADAPTIVE ATTACK

In this subsection, we provide an analysis of Subspace Detection against adaptive attacks, i.e., the attacker is aware of our detection strategy, thus attempting to evade our detection.

### F.2.1 EXPERIMENTAL SETTING

**Adaptive attack.** We designed an adaptive attack scenario based on the Blended attack to evaluate our proposed method. In this setting, we assume the attacker has partial to full knowledge of the text transformations employed by our method. Specifically, the attacker is aware that our detection method may use three types of transformations: Descriptive Rephrasing, Font Styling, and Language Translation. To evade detection, the attacker adaptively crafts poisoned samples. They select 2000 out of 5000 poisoned samples and replace their corresponding target text captions with target text transformed by these three transformation types. For systematic evaluation, we simulated seven attack scenarios, corresponding to the attacker being aware of each of the seven possible combinations of these transformations, *i.e.*, *Font, Language, Description, Font+Language, Font+Description, Language+Description, and Language+Description+Font*. We categorize the seven attack scenarios into two cases:

- **Case 1: The Attacker Possesses Incomplete Knowledge.** This case, corresponding to the first six scenarios, reflects a more realistic threat model where the attacker cannot fully know the defender

employed transformations (*e.g.*, being unaware that Descriptive Rephrasing is used). Furthermore, a realistic attacker would also have incomplete knowledge of the transformation parameters (*e.g.*, not knowing the specific font used for Font Styling). To construct a more challenging evaluation and test our method against a stronger adversary, we assume that for the subset of transformation types the attacker is aware of, they also know the exact implementation parameters.

- **Case 2: The Attacker Possesses Complete Knowledge.** This last scenario represents the most severe threat model. We assume the attacker has full knowledge: they are aware of all three transformations used by the defender and also know the exact parameters for each.

**Detection.** Throughout all these adaptive attack experiments, our detection method consistently utilizes all three types of transformations to approximate the local text manifold.

### F.2.2 EXPERIMENTAL RESULTS

The results of our adaptive attack experiments are summarized in the Tab. 9, showing the AUROC and F1-score of our detection method as the attacker incorporates more transformations.

Table 9: Evaluation of Subspace Detection against adaptive attacks. Each column represents a different attack scenario, defined by the set of text transformations the attacker is aware of and uses for poisoning.

| | Case 1 | | | | | | Case 2 |
| --- | --- | --- | --- | --- | --- | --- | --- |
| | Font | Language | Description | Font+Language | Font+Description | Language+Description | Language+Description+Font |
| AUROC | 0.8833 | 0.9536 | 0.9602 | 0.8776 | 0.8571 | 0.8036 | 0.6559 |
| F1-score | 0.8909 | 0.9515 | 0.9557 | 0.8656 | 0.8557 | 0.8075 | 0.6922 |

As Tab. 9 indicates, the detection performance shows a downward trend as the attacker gains more knowledge of the defender's transformations. This is an expected result, as the attacker can more effectively craft poisoned samples to align with the text manifold used for detection. We would like to highlight two key takeaways from these results:

**Effectiveness in Case 1.** In realistic scenarios (Case 1), it is highly impossible for an attacker to gain complete knowledge of text transformations used by the defender. Our results show that when the attacker has only partial knowledge, our method maintains a satisfactory AUROC and F1-score, demonstrating its effectiveness in realistic adaptive threat models.

**Inherent Resilience in in Case 2.** Even in the most severe scenario (Case 2) where the attacker knows all three transformations and their parameters, our defense does not fail completely, maintaining an AUROC of 0.6559. This resilience stems from a key difference: the attacker is limited to poisoning the dataset with a limited set of text variants, whereas our method defends by sampling a diverse set of text features from a continuous approximated local text manifold. Thus, the attacker's poisoned samples cannot achieve perfect alignment with the manifold. Our manifold-based sampling ensures there will always be sampled text features that can distinguish the poisoned images, thus preserving the detection capability.

### F.3 DISCUSSION ON THE CHOICE OF DISTANCE METRIC

In our proposed Subspace Detection, we employ the Euclidean distance, denoted as $d_2(\mathbf{z}_v, \mathbf{z}_d) = \|\mathbf{z}_v - \mathbf{z}_d\|_2$, to measure the deviation of a test image feature from the sampled text features in the region-of-interest in Eq. 1. This choice is grounded in both theoretical consistency with our manifold approximation strategy and empirical performance.

**Theoretical Consistency with PCA and Gaussian Modeling.** First, our method utilizes PCA to construct the low-dimensional subspace $\mathcal{S}$ that approximates the local text manifold. A fundamental property of PCA is that it finds the projection minimizing the reconstruction error under the $L_2$ norm. Thus, our linear approximation of the manifold is optimal in an $L_2$ sense. By using the Euclidean distance to measure the deviation of an image feature from this PCA-constructed manifold, we ensure the construction of the subspace and the deviation measure used for detection are based on the same

Table 10: Comparison using $L_1$ vs. $L_2$ distance metrics on ImageNet-1K dataset with ResNet-50.

| Attack | BadNet | | Blended | | SIG | | WaNet | | TrojVQA | | Carlini&Terzis | | BadCLIP | |
|---|---|---|---|---|---|---|---|---|---|---|---|---|---|---|
| Metric↓ | AUROC | F1-score | AUROC | F1-score | AUROC | F1-score | AUROC | F1-score | AUROC | F1-score | AUROC | F1-score | AUROC | F1-score |
| $L_1$ | 0.671 | 0.695 | 0.693 | 0.681 | 0.525 | 0.533 | 0.872 | 0.923 | 0.738 | 0.827 | 0.512 | 0.497 | 0.490 | 0.532 |
| $L_2$ | 0.962 | 0.920 | 0.982 | 0.969 | 0.692 | 0.788 | 0.931 | 0.901 | 0.925 | 0.879 | 0.994 | 0.968 | 0.966 | 0.963 |

underlying geometry, avoiding a mismatch between how the manifold is learned and how distances to it are measured.

Second, we characterize the discriminative region-of-interest within local text manifold using Gaussian distributions. This assumption implies a local Gaussian structure for text feature variations. Given that clean images are well-aligned with the region-of-interest in the CLIP embedding space, the feature of a clean image $\mathbf{z}_v$ should also adhere to this local geometry. The difference between the image feature $\mathbf{z}_v$ and a text feature $\mathbf{z}_d$ on the manifold can be modeled as a small perturbation. The residual, $\delta = \mathbf{z}_v - \mathbf{z}_d$, can be viewed as approximately Gaussian noise around the local text manifold, i.e., $\delta \sim \mathcal{N}(\mathbf{0}, \sigma^2 \mathbf{I})$. Furthermore, poisoned image features either have a much larger variance or a bias that pushes them away from the same region. The negative log-likelihood of $\delta$ is then proportional to $\|\delta\|_2^2$, which is the squared Euclidean distance. Therefore, thresholding $d_2(\mathbf{z}_v, \mathbf{z}_d)$ is equivalent to using a sufficient statistic for distinguishing high-likelihood (near-manifold, i.e., clean) samples from low-likelihood (off-manifold, i.e., poisoned) samples. **Empirical Comparison with $L_1$ Distance.** In Tab 10, we experiment with replacing the $L_2$ distance by $L_1$ in Subspace Detection. This modification consistently led to noticeably worse detection performance across multiple attacks. This is because $L_1$ is no longer aligned with (i) the PCA subspace construction and (ii) the Gaussian-based modeling of region-of-interest discussed above.

## F.4 COMPUTATIONAL COST

Table 11: Comparison of per-image CLIP inference time and test-time detection runtime (in seconds) for different backdoor detection methods on the same hardware. Results are based on ImageNet-1K.

| Detection Method | CLIP inference time | SCALE-UP | STRIP | Description | Font | Language | Subspace Detection |
|---|---|---|---|---|---|---|---|
| **Runtime (s)** | 0.0086 | 0.016 | 0.057 | 0.011 | 0.012 | 0.011 | 0.009 |

A key advantage of our Subspace Detection is its time efficiency at test time. The core of our approach is fitting a uniform mixture of Gaussians, which characterizes the region-of-interest for a given semantic concept. Crucially, this distribution is label-specific, not instance-specific. Therefore, the distribution for each class can be pre-computed once and then reused for every subsequent test image predicted to be in that class. This effectively decouples the computational cost into two phases: *a one-time and offline setup cost*, and *a highly efficient online detection cost*.

**Offline setup cost.** The offline setup cost encompasses the characterization of the region-of-interest for each class (corresponding to Sec. 4.2.1). The specific steps include: *(1) generating semantic text variants, (2) approximating the local manifold by fitting a low-dimensional subspace, (3) characterizing the region-of-interest within subspace by sampling along positive directions. (4) fitting the uniform mixture of Gaussians to model this region.* On a server equipped with a 2.6 GHz Intel Xeon Gold 6278C CPU, the characterization for a single class takes approximately 0.1963 seconds. This demonstrates that the per-class setup cost is minimal, making it feasible to pre-compute distributions for all classes in a downstream dataset before deployment.

**Online detection cost.** The primary benefit of this offline pre-computation is the resulting speed of online detection. At test time, the detection process for a single test image only requires: *(1) running CLIP inference to obtain its predicted label and the corresponding image feature, (2) loading the pre-computed mixture of Gaussians distribution corresponding to the image's predicted class and sampling text features from this distribution, (3) calculating their average Euclidean distance to the image feature.* To validate the efficiency of online detection, we compare the per-image online detection runtime of Subspace Detection against other compared methods. As shown in Tab. 11, our method exhibits a significant time efficiency, which is critical for real-world deployments where low-latency inference is a critical requirement.

### F.5 Comparisons with the BDetCLIP

As shown in Tab. 12, we present the detection results of BDetCLIP and Subspace Detection on three different datasets. Specifically, ImageNet-R and ImageNet-Sketch are more challenging variants of the ImageNet-1K dataset. The results indicate that although the detection performance of Subspace Detection and BDetCLIP on ImageNet-R and ImageNet-Sketch both decrease slightly compared to their performance on ImageNet-1K, Subspace Detection demonstrates slightly higher effectiveness than BDetCLIP.

Table 12: Comparison with the BDetCLIP detection on three datasets on ResNet-50.

| Detection | Dataset→ Attack↓ | ImageNet-1K | | ImageNet-R | | ImageNet-Sketch | |
|---|---|---|---|---|---|---|---|
| | | AUROC | F1-score | AUROC | F1-score | AUROC | F1-score |
| BDetCLIP | Blended | 0.957 | 0.915 | 0.813 | 0.748 | 0.727 | 0.755 |
| | WaNet | 0.950 | 0.909 | 0.702 | 0.685 | 0.704 | 0.713 |
| | TrojVQA | 0.907 | 0.899 | 0.834 | 0.718 | 0.786 | 0.740 |
| | Average | 0.938 | 0.908 | 0.783 | 0.717 | 0.739 | 0.736 |
| Subspace Detection | Blended | 0.982 | 0.969 | 0.884 | 0.919 | 0.890 | 0.916 |
| | WaNet | 0.931 | 0.901 | 0.826 | 0.858 | 0.882 | 0.859 |
| | TrojVQA | 0.925 | 0.879 | 0.865 | 0.856 | 0.863 | 0.854 |
| | Average | 0.946 | 0.916 | 0.858 | 0.878 | 0.878 | 0.876 |

## G Limitation

While our method demonstrates strong detection performance, there is a limitation that requires further improvement in future work. A key component of Subspace Detection is to generate text variants to approximate the local text manifold corresponding to a specific concept. In the current implementation, generating variants, particularly via descriptive rephrasing and language translation, requires the involvement of LLMs. While effective, this process incurs API expenses, potentially limiting scalability in practice. In future work, we plan to integrate external translation systems or design models for generating descriptions, which could reduce costs and improve the practical generality of Subspace Detection.

## H The Use of Large Language Models

In this work, we utilized the LLM in two aspects: as a component within our method and as a writing assistant. In our method, we approximate the local text manifold using various variants of the original text. To generate these variants, we use LLM to implement text transformations on the original one, such as language translation and descriptive rephrasing. Additionally, we used the LLM to refine the manuscript's language, improving its clarity and grammar. All scientific contributions, such as the core research ideas and experimental design, are the original work of the human authors.

