# OpenReview forum: "Test-Time Poisoned Sample Detection by Exploiting Shallow Malicious Matching in Backdoored CLIP"
_ICLR.cc/2026/Conference — ICLR 2026 Poster_

### Official Review · Reviewer_PUEo · 2025-10-30

**Soundness:** 3
**Presentation:** 3
**Contribution:** 3
**Rating:** 6
**Confidence:** 4

**Summary:**

This paper finds that images with a hidden backdoor ("poisoned" images) in a compromised CLIP model show a detectable flaw called "shallow malicious matching." While these images are forced to match a specific target text caption, they do not align with the broader meaning of that caption. In contrast, normal "benign" images show "deep benign matching," meaning their features are close to the whole group of text features that share the same meaning. Using this discovery, the authors propose "Subspace Detection," a method to spot poisoned images during testing. It works by checking if an image's feature is far away from the semantic neighborhood of its predicted text; a large distance indicates a poisoned image. Experiments show this method works better than existing defenses across different attacks and datasets.

**Strengths:**

1. Novelty. The paper's core insight is somewhat original. The concepts of "shallow malicious matching" versus "deep benign matching" offer a novel and intuitive lens through which to understand and detect backdoor attacks in vision-language models. The proposed "Subspace Detection" method, which leverages the geometry of the text manifold, is a creative and non-obvious application of this insight.

2. Clarity: The paper is well-written and structured. The central concept is introduced clearly in the abstract and introduction, and the illustrative Figure 1 effectively reinforces the explanation of "shallow" vs. "deep" matching. The description of the three-step "Subspace Detection" method is logical and easy to follow, making the technical contribution accessible even to readers not deeply versed in backdoor literature.

**Weaknesses:**

1. The paper should compare its method more directly with BDetCLIP. A clearer side-by-side comparison would help readers understand the specific improvements and differences.

2. Testing the method against more types of advanced multimodal backdoor attacks (beyond BadCLIP) would make the results stronger.

3. It would be good to test if the idea works on other multimodal models, not just CLIP. If it does, it would show that the finding is more general and important.

**Questions:**

What is the underlying cause of shallow malicious matching? Could you elaborate on the mechanism behind shallow malicious matching?

---

> ### Author Response · Authors · 2025-11-22
>
> **Q1: The paper should compare its method more directly with BDetCLIP. A clearer side-by-side comparison would help readers understand the specific improvements and differences.**
>
> R1: We thank the reviewer for this comment for the comparison between our Subspace Detection and BDetCLIP. Below we summarize the methodological differences and provide an experimental comparison.
>
> (1) Methodological comparison
> The core differences between Subspace Detection and BDetCLIP stem from their distinct motivations, which in turn lead to different text transformation strategies and detection metrics. We summarize these differences in the table below, followed by a detailed discussion.
>
> | Aspect | BDetCLIP | Subspace Detection |
> | :--- | :--- | :--- |
> | **Motivation** | Based on the **insensitivity** of poisoned images to semantic changes. The alignment between a poisoned image and its target label text is so strong that it persists even when the text is perturbed with random, irrelevant information. However, benign images are sensitive to such changes. | Based on the **fragility** of backdoor alignments, which we term **shallow malicious matching.** The connection between a trigger and a target label is superficial and does not generalize to semantically equivalent variants of the text. However, benign images exhibit **deep benign matching,** which is robust to these variations. |
> | **Text Transformation Strategy** | Generates two sets of prompts: 1. **Benign Prompts:** Fine-grained, attribute-based descriptions that are semantically consistent with the target label. 2. **Malignant Prompts:** The original class template concatenated with a random, semantically unrelated sentence. | Generates **semantically equivalent variants** of the predicted text caption through transformations that preserve meaning, such as: 1. **Descriptive Rephrasing** 2. **Font Styling** 3. **Language Translation**|
> | **Text Transformation Demonstration**  Class: goldfish |1. **Benign Prompts:** Goldfish are known for their bright orange or gold color but they can also come in other colors like white, black, red, and yellow. 2. **Malignant Prompts:** A photo of a goldfish. The bright sun cast shadows on the bustling city street. *(Both prompt types explicitly retain the target class name.)*| 1. **Descriptive Rephrasing:** A small, ornamental fish, known for its bright golden hue and delicate, flowing fins. 2. **Font Styling** $\scr{goldfish}$ 3. **Language Translation:** سمكة ذهبية  *(Descriptive rephrasing often results in text that does not contain the original target class name.)* |
> | **Detection Metric** | Measures the **sensitivity** of an image to semantic changes by calculating the difference in cosine similarity between the image and the benign vs. malignant prompts. A small difference indicates insensitivity, flagging the image as poisoned. | Measures the **deviation** of an image feature from a discriminative region on the local text manifold. This involves: 1. Approximating the local text manifold with a low-dimensional subspace. 2. Probing a region-of-interest within the subspace that maximally separates benign and poisoned samples. 3. Calculating the Euclidean distance of the image feature to this region. A large distance indicates a deviation, flagging the image as poisoned. |
>
> As outlined in the table, the two methods are different:
> * **On Motivation**: The core hypotheses are fundamentally different. BDetCLIP hypothesizes that the backdoor learning process creates an abnormally robust but incorrect alignment that is insensitive to semantic disruption. In contrast, our work identifies that this alignment is actually extremely fragile and superficial, failing to generalize to minor semantic-preserving textual changes.
> * **On Text Transformation**: The different motivations naturally lead to different text transformation strategies. BDetCLIP creates a semantic perturbation by injecting random and unrelated text to test for insensitivity. Crucially, its prompts always include the target class name. Subspace Detection generates semantically preserved variants to test for generalization. Our descriptive rephrasing, for instance, describes the concept without using the target class name, directly challenging the model to demonstrate a deeper semantic understanding beyond simple keyword matching.
> * **On Detection Metric**: This is where the most significant technical innovation of our work lies. BDetCLIP's metric is a direct comparison of similarity scores derived from a few GPT-generated prompts. Instead of simply relying on handcrafted text variants, Subspace Detection approximates the corresponding local text manifold constructed from them. It then identify and sample from a highly discriminative region-of-interest within it that maximally amplifies the distance between benign and poisoned image features. By measuring an image's deviation from this region, our metric provides a much more robust and effective signal for detection.

---

> ### Author Response · Authors · 2025-11-22
>
> In summary, while both methods are test-time poisoned image detection, BDetCLIP tests for insensitivity to semantic changes, whereas Subspace Detection tests for the lack of generalization to semantically equivalent variants. Our method's foundation on the "shallow vs. deep matching" hypothesis and its well-designed manifold-based metric represent a totally different method.
>
> (2) Experimental comparison
> * **Experimental setting:** The evaluations are conducted on the validation sets of ImageNet-1K, ImageNet-R, and ImageNet-Sketch datasets under three backdoor attacks.
> * **Experimental results:** The detection results of our method demonstrate competitive performance with BDetCLIP, showing its effectiveness.
>
> |Detection|Dataset→|ImageNet-1K|ImageNet-R|ImageNet-Sketch|
> |-|-|-|-|-|
> ||Attack↓| AUROC/F1-score | AUROC/F1-score|AUROC/F1-score|
> |BDetCLIP|Blended|0.957/0.915| 0.813/0.748|0.727/0.755|
> ||WaNet|0.950/ 0.909|0.702/0.685|0.704/0.713|
> ||TrojVQA|0.907/0.899|0.834/0.718|0.786/0.740|
> ||**Average**|**0.938/0.908**|**0.783/0.717**|**0.739/0.736**|
> |SubSpace Detection|Blended|0.982/0.969|0.884/0.919|0.890/0.916|
> ||WaNet|0.931/0.901|0.826/0.858|0.882/0.859|
> ||TrojVQA|0.925/0.879|0.865/0.856|0.863/0.854|
> || **Average**|**0.946/0.916**|**0.858/0.878**|**0.878/0.876**|

---

> ### Author Response · Authors · 2025-11-22
>
> **Q2: Testing the method against more types of advanced multimodal backdoor attacks (beyond BadCLIP) would make the results stronger.**
>
> R2: Thanks for the comment. We have added anothor advanced multimodal backdoor attack, also named BadCLIP[2], to evaluate our method. Our method achieves an AUROC of 0.934 and an F1-score of 0.921, demonstrating its effectiveness.
>
> [2] Bai J, Gao K, Min S, et al. BadCLIP: Trigger-Aware Prompt Learning for Backdoor Attacks on CLIP[C]//2024 IEEE/CVF Conference on Computer Vision and Pattern Recognition (CVPR). IEEE, 2024: 24239-24250.
>
> **Q3: It would be good to test if the idea works on other multimodal models, not just CLIP. If it does, it would show that the finding is more general and important.**
>
> R3: Thank you for the comment. Unlike CLIP, OpenCLIP uses a larger-scale pre-training dataset, LAION, and also incorporates various advanced techniques, such as utilizing a pre-trained language model and scaling the model architecture to a larger size. Therefore, we deploy the BadCLIP attack method on OpenCLIP to further evaluate our approach. The results show an AUROC of 0.978 and an F1-score of 0.974, demonstrating the generalizability of our method.

---

> ### Author Response · Authors · 2025-11-22
>
> **Q4: What is the underlying cause of shallow malicious matching? Could you elaborate on the mechanism behind shallow malicious matching?**
>
> R4: We thank the reviewer for this comment. Below we first give the intuition behind shallow malicious matching, then we develop a theoretical model which formalizes this phenomenon and explains why it arises.
>
> **(1) Intuitive Explanation**
>
> * Deep Benign Matching: The robustness of matching for benign images **stems from CLIP's large-scale pre-training**. This process enables the model to learn rich visual concepts deeply connected to the text manifold of associated semantic concept. Consequently, the matching for clean images is robust to semantic-preserving text modifications.
> * Shallow Malicious Matching: In contrast, **backdoor learning forces a spurious correlation** between a trigger pattern and a specific target text, which often contradicts CLIP's foundational knowledge. This creates a fragile link where the poisoned image's malicious matching with the target text hardly generalizes to semantically equivalent variants of that text. Any minor modification to the target text can break this superficial connection.
>
> **(2) Theoretical Analysis**
>
> Let $\hat{f}_ v$ and $\hat{f}_ t$ be the backdoored CLIP image and text encoders, respectively, and denote image and text features by $\mathbf{z}_ v=\hat{f}_ v(\mathbf{v})$ and $\mathbf{z}_ t=\hat{f}_ t(\mathbf{t}) \in \mathbb{R}^d$. To formalize the intuition above, we first model the local text manifold corresponding to a semantic concept. For a concept $c$, we approximate the corresponding local text manifold of $c$ by the affine subspace:
>
> $S_ c:= \left\\{ \mathbf{\mu}_ c+U_ c \mathbf{\alpha}: \mathbf{\alpha} \in \mathbb{R}^k \right\\},$
>
> where $\mathbf{\mu}_c$ is the center, columns of $U_c \in \mathbb{R}^{d \times k}$ with $U_c^{\top} U_c=\mathbf{I}_k$ span this subspace.
>
>
> **Lemma 1.** Let $P_c:=U_c U_c^{\top}$ denote the projection matrix onto $S_c$. For any image feature $z_v$, its distance to the text manifold is
>
> $
> \operatorname{dis}\left(\mathbf{z}_v, S_c\right):=\min _\alpha\left\|\|\mathbf{z}_v-\left(\mathbf{\mu}_c+U_c \alpha\right)\right\|\|=\left\|\|\left(\mathbf{I}-P_c\right)\left(\mathbf{z}_v-\mathbf{\mu}_c\right)\right\|\| .
> $
>
> **Remark**: The term $(\mathbf{I} - P_c)(\mathbf{z}_v - \mathbf{\mu}_c)$ represents the residual part of the image feature that cannot be explained by the principal semantic directions of concept $c$. Thus, $\operatorname{dis}(\mathbf{z}_v, S_c)$ quantifies the semantic gap between the image and the concept manifold.
>
>
> **Proposition 1 (Existence of shallow malicious matching features).**  Let the local text manifold of the target concept $c^{\star}$ be approximated by an affine subspace $S_ {c^{\star}}=\left\\{\mathbf{\mu}_ {c^{\star}}+U_ {c^{\star}} \mathbf{\alpha}: \mathbf{\alpha} \in \mathbb{R}^k\right\\}$, where $\mathbf{\mu}_ {c^{\star}} \in \mathbb{R}^d$ and the columns of $U_ {c^{\star}} \in \mathbb{R}^{d \times k}$ form an orthonormal basis. Let the target text feature $\mathbf{z}_ t^{\star}=\hat{f}(\mathbf{t^{\star}}) \in \mathbb{R}^d$ lie on this manifold, i.e., $\mathbf{z}_ t^{\star} \in S_{c^{\star}}$, and denote its norm by $a:=\left\|\|\mathbf{z}_ t^{\star}\right\|\|>0$. Assume that for all benign images of their corresponding concept $c$, their image features $\mathbf{z}_ v^{\text {benign }}=\hat{f}(\mathbf{v}^{\text {benign}})$ satisfy $\operatorname{dis}\left(\mathbf{z}_ v^{\text {benign}}, S_ {c}\right) \leq \delta_ {\text {benign}}$ for some small $\delta_ {\text {benign }}>0$. For any $\varepsilon \in(0,1)$ satisfying
>
> $
> \delta_{\text {benign }}<a \sqrt{\frac{1}{(1-\varepsilon)^2}-1}, \quad (A)
> $
>
> there exists a feature vector $\mathbf{z}_v^{\text {poison}}=\hat{f}(\mathbf{v}^{\text{poison}}) \in \mathbb{R}^d$ such that
>
> $
> \cos \left(\mathbf{z}_ v^{\text {poison}}, \mathbf{z}_ t^{\star}\right) \geq 1-\varepsilon \quad \text {and} \quad \operatorname{dis}\left(\mathbf{z}_ v^{\text {poison}}, S_ {c^{\star}}\right)>\delta_ {\text {benign}}.
> $
>
> Moreover, if $\delta_ {\text {benign }}$ is sufficiently small compared to $a$, the distance $\operatorname{dis}\left(\mathbf{z}_ v^{\text {poison }}, S_ {c^{\star}}\right)$ can be larger than $\delta_ {\text {benign}}$, i.e., poisoned features can be almost perfectly aligned with the target text in cosine similarity while remaining far away from the target text manifold in the Euclidean distance, which is what we call shallow malicious matching.

---

> ### Author Response · Authors · 2025-11-22
>
> **Proof:** Since $\mathbf{z}_ t^{\star} \in S_{c^{\star}}$, there exists $\mathbf{\alpha}_ 0$ such that
>
> $
> \mathbf{z}_ t^{\star}=\mathbf{\mu}_ {c^{\star}}+U_ {c^{\star}} \mathbf{\alpha}_ 0 .
> $
>
>
> Equivalently, we have $\tilde{\mathbf{w}}:=\mathbf{z}_ t^{\star}-\mathbf{\mu}_ {c^{\star}} \in \operatorname{span}\left(U_ {c^{\star}}\right)$.
> Consider the orthogonal complement $\operatorname{span}\left(U_{c^{\star}}\right)^{\perp}$. For an arbitrary scalar $c>0$, choose any vector
>
> $
> \mathbf{u}_ {\perp} \in \operatorname{span}\left(U_ {c^{\star}}\right)^{\perp} \quad \text { with } \quad\left\|\|\mathbf{u}_ {\perp}\right\|\|=c .
> $
>
>
> We consider a simple construction of $\mathbf{z}_v^{\text{poison}}$:
>
> $
> \mathbf{z}_ v^{\text{poison}}:=\mathbf{z}_ t^{\star}+\mathbf{u}_ {\perp} .
> $
>
> **Cosine similarity with the target text.**
> Since $\left(\mathbf{z}_ t^{\star}\right)^{\top} \mathbf{u}_ {\perp}=0$, we have
>
> $
> \left\|\left\|\mathbf{z}_ v^{\text {poison }}\right\|\right\|^2=\left\|\left\|\mathbf{z}_ t^{\star}+\mathbf{u}_ {\perp}\right\|\right\|^2=\left\|\left\|\mathbf{z}_ t^{\star}\right\|\right\|^2+\left\|\left\|\mathbf{u}_ {\perp}\right\|\right\|^2=a^2+c^2,
> $
>
> and
>
> $
> \left(\mathbf{z}_ v^{\text {poison }}\right)^{\top} \mathbf{z}_ t^{\star}=\left(\mathbf{z}_ t^{\star}+\mathbf{u}_ {\perp}\right)^{\top} \mathbf{z}_ t^{\star}=\left\|\|\mathbf{z}_ t^{\star}\right\|\|^2=a^2 .
> $
>
>
> The cosine similarity between $\mathbf{z}_v^{\text {poison }}$ and $\mathbf{z}_t^{\star}$ is
>
> $
> \cos \left({\mathbf{z}}_v^{\text {poison}}, {\mathbf{z}}_t^{\star}\right)=\frac{\left(\mathbf{z}_v^{\text {poison}}\right)^{\top} \mathbf{z}_t^{\star}}{\left\|\left\|\mathbf{z}_v^{\text {poison}}\right\|\right\|\left\|\left\|\mathbf{z}_t^{\star}\right\|\right\|}=\frac{a^2}{\sqrt{a^2+c^2} a}=\frac{a}{\sqrt{a^2+c^2}}=\frac{1}{\sqrt{1+\frac{c^2}{a^2}}}.
> $
>
>
> Given any $\varepsilon \in(0,1)$, we want
>
> $
> \cos \left(\mathbf{z}_v^{\text {poison }}, \mathbf{z}_t^{\star}\right) \geq 1-\varepsilon,
> $
>
> this is equivalent to choose $c$ such that
>
> $
> c \leq a \sqrt{\frac{1}{(1-\varepsilon)^2}-1} .
> $
>
> **Distance to the text manifold.**
>
> By construction, we have
>
> $
> \mathbf{z}_ v^{\text {poison }}-\mathbf{\mu}_ {c^{\star}}=\left(\mathbf{z}_ t^{\star}-\mathbf{\mu}_ {c^{\star}}\right)+\mathbf{u}_ {\perp}=\tilde{\mathbf{w}}+\mathbf{u}_ {\perp}.
> $
>
> Since $\tilde{\mathbf{w}} \in \operatorname{span}\left(U_ {c^{\star}}\right)$ and $\mathbf{u}_ {\perp} \in \operatorname{span}\left(U_ {c^{\star}}\right)^{\perp}$, by the definition of $\operatorname{dis}\left(\cdot, S_ {c^{\star}}\right)$ in Lemma 1, we have
>
> $
> \operatorname{dis}\left(\mathbf{z}_ v^{\text {poison }}, S_ {c^{\star}}\right)=\left\|\left\|\left(I-U_ {c^{\star}} U_ {c^{\star}}^{\top}\right)\left(\mathbf{z}_ v^{\text {poison }}-\mathbf{\mu}_ {c^{\star}}\right)\right\|\right\|=\left\|\left\|\left(I-U_ {c^{\star}} U_ {c^{\star}}^{\top}\right)\left(\tilde{\mathbf{w}}+\mathbf{u}_ {\perp}\right)\right\|\right\|=\left\|\left\|\mathbf{0}+\mathbf{u}_ {\perp}\right\|\right\|=c .
> $
>
> Hence, if we choose $c>\delta_{\text {benign }}$, then
>
> $
> \operatorname{dis}\left(\mathbf{z}_ v^{\text {poison }}, S_ {c^{\star}}\right)=c>\delta_ {\text {benign }},
> $
>
> i.e., the poisoned image feature lies strictly farther away from the target manifold than any benign target-class image.
>
> **Combining the two constraints.**
> To simultaneously have
>
> $
> \operatorname{dis}\left(\mathbf{z}_ v^{\text {poison }}, S_ {c^{\star}}\right)=c>\delta_ {\text {benign }} \quad \text { and } \quad \cos \left(\mathbf{z}_ v^{\text {poison }}, \mathbf{z}_ t^{\star}\right) =\frac{1}{\sqrt{1+\frac{c^2}{a^2}}} \geq 1-\varepsilon,
> $
>
> we need a $c$ satisfying
>
> $
> \delta_{\text {benign }}<c \leq a \sqrt{\frac{1}{(1-\varepsilon)^2}-1} .
> $
>
>
> By assumption (A), such a $c$ exists. For any $c$ in this interval, picking any $\mathbf{u}_ {\perp}$ with $\left\|\left\|\mathbf{u}_ {\perp}\right\|\right\|=c$ in $\operatorname{span}\left(U_{c^{\star}}\right)^{\perp}$ and defining $\mathbf{z}_ v^{\text {poison}}= \mathbf{z}_ t^{\star}+\mathbf{u}_ {\perp}$ yields a feature that satisfies both inequalities.
>
> Moreover, if $\delta_{\text {benign }}$ is sufficiently small compared to $a$, the upper bound $a \sqrt{\frac{1}{(1-\varepsilon)^2}-1}$ is much larger than $\delta_{\text {benign}}$, so we can select $c$ with $c \gg \delta_{\text {benign}}$, making the gap $\operatorname{dis}\left(\mathbf{z}_ v^{\text {poison}}, S_ {c^{\star}}\right) / \delta_ {\text {benign}}$ large while keeping $\cos \left(\mathbf{z}_ v^{\text {poison}}, \mathbf{z}_ t^{\star}\right) \geq 1-\varepsilon$. $\square$

---

> ### Author Response · Authors · 2025-11-22
>
> **Remark:** in Proposition 1 we assume that for all benign images of their corresponding concept $c$, their image features $\mathbf{z}_ v^{\text {benign }}=\hat{f}\left(\mathbf{v}^{\text {benign }}\right)$ satisfy $\operatorname{dis}(\mathbf{z}_ v^{\text{benign}}, S_ {c}) \le \delta_ {\text{benign}}$ for some small $\delta_{\text {benign }}>0$. This assumption over deep benign matching naturally stems from CLIP's large-scale pre-training, where benign images of concept $c$ are well-aligned with a great number textual features $\left\\{\mathbf{z}_ t^{(j)}\right\\}_ {j=1}^m$ in $S_ {c}$ which approxiates the local text manifold of $c$. The goal for a benign image feature is to maximize the average cosine:
>
> $
> \frac{1}{m} \sum_ {j=1}^m \cos \left(\mathbf{z}_ v^{\text{benign}}, \mathbf{z}_ t^{(j)}\right)=\bar{\mathbf{\mu}}_ {c}^{\top} \bar{\mathbf{z}}_ v^{\text{benign}},
> $
>
> where $\bar{\mathbf{z}}_ v^{\text{benign}}={\mathbf{z}}_ v^{\text{benign}}/\|\|{\mathbf{z}}_ v^{\text{benign}}\|\|$, $\bar{\mathbf{\mu}}_ {c}=\frac{1}{m} \sum_ j \bar{\mathbf{z}}_ t^{(j)}/\left\|\|\bar{\mathbf{z}}_ t^{(j)}\right\|\| \in S_ {c}$. It's easy to find that The optimal $\bar{\mathbf{z}}_ v^{\text{benign}}$ is $\bar{\mathbf{\mu}}_ {c}/\|\|\bar{\mathbf{\mu}}_ {c}\|\|$, which lies in $S_ {c}$, thus its distance to $S_{c}$ is an extramely small $\delta_ {\text {benign }}$.
>
> In contrast, Proposition 1 shows that there exist backdoored features $z_v^{\text{poison}}$ which is aligned with the target caption in cosine similarity, while their distance to $S_ {c^{\star}}$ in the pre-normalization space is larger than $\delta_ {\text{benign}}$. This is what we called shallow malicious matching. Our Subspace Detection leverages precisely this geometric seperation to separate poisoned image features from deeply matched benign ones.

---

### Official Review · Reviewer_Zp8v · 2025-11-01

**Soundness:** 3
**Presentation:** 3
**Contribution:** 2
**Rating:** 4
**Confidence:** 4

**Summary:**

The paper proposes Subspace Detection, a test-time detection of poisoned images in backdoored CLIP models. The key observation is that clean images align deeply with the local semantic manifold of their class text representations, while poisoned images only align superficially to the target caption and deviate from semantically equivalent variants. Based on this, the method measures the image’s average distance to sampled text features for poison detection.

**Strengths:**

1. The identification of the contrast between shallow malicious matching and deep benign matching offers a useful perspective on how poisoned images behave in CLIP embedding space.

2. The proposed subspace-based detection framework is conceptually clear and methodologically coherent.

3. The method works at test-time and does not require model retraining or access to original training data, which increases practical applicability.

**Weaknesses:**

1. Not enough baseline for comparisons. Methods like [1] should be included in baseline.

2. The transformation ablation shows “more transforms help,” but the paper doesn’t analyze why some single transforms (e.g., description vs. font) differ so much, nor the minimal set needed to reach near-max performance. A short analysis would help the clarifications.

3. Thresholding is set via a clean reference set; sensitivity to the clean-set distribution and false-positive trade-offs is not fully explored.

4. Margins between clean and poisoned can be small, making the differences between shallow vs deep matching not as significant.

5. Lack of theoretical supports for the shallow malicious matching vs deep benign matching.

6.The method is not evaluated against label-consistent backdoor attacks or cases where poisoned images match multiple semantically similar target prompts. It is unclear whether the shallow-vs-deep matching assumption holds in these scenarios.


[1] Niu, Yuwei, et al. "Test-Time Multimodal Backdoor Detection by Contrastive Prompting." Forty-second International Conference on Machine Learning.

**Questions:**

See weaknesses.

---

> ### Author Response · Authors · 2025-11-23
>
> **Q1: Not enough baseline for comparisons. Methods like [1] should be included in baseline.**
>
> [1] Niu, Yuwei, et al. "Test-Time Multimodal Backdoor Detection by Contrastive Prompting." Forty-second International Conference on Machine Learning.
>
> R1: Thanks for the suggestion. We have added the BDetCLIP[1] as a baseline in our experiments, as explained below:
>
> * **Experimental setting:** The evaluations are conducted on the validation sets of ImageNet-1K, ImageNet-R, and ImageNet-Sketch datasets under three backdoor attacks.
> * **Experimental results:** The detection results of our method demonstrate competitive performance with BDetCLIP, showing its effectiveness.
>
> |Detection|Dataset→|ImageNet-1K|ImageNet-R|ImageNet-Sketch|
> |-|-|-|-|-|
> ||Attack↓| AUROC/F1-score | AUROC/F1-score|AUROC/F1-score|
> |BDetCLIP|Blended|0.957/0.915| 0.813/0.748|0.727/0.755|
> ||WaNet|0.950/ 0.909|0.702/0.685|0.704/0.713|
> ||TrojVQA|0.907/0.899|0.834/0.718|0.786/0.740|
> ||**Average**|**0.938/0.908**|**0.783/0.717**|**0.739/0.736**|
> |SubSpace Detection|Blended|0.982/0.969|0.884/0.919|0.890/0.916|
> ||WaNet|0.931/0.901|0.826/0.858|0.882/0.859|
> ||TrojVQA|0.925/0.879|0.865/0.856|0.863/0.854|
> || **Average**|**0.946/0.916**|**0.858/0.878**|**0.878/0.876**|
>
> **Q2: The transformation ablation shows “more transforms help,” but the paper doesn’t analyze why some single transforms (e.g., description vs. font) differ so much, nor the minimal set needed to reach near-max performance. A short analysis would help the clarifications.**
>
> R2: Thanks for the comment.
> * **This is because attackers typically use a description containing the target label as the caption, which facilitates the alignment between the target text description and the poisoned images, leading to a decrease in the effectiveness of the Description-only method.** However, the poisoned images lack such alignment with the target label in other transformations, such as Language and Font. To better analyze this phenomenon, we also take BadCLIP as an example and calculate the difference in average Euclidean distance between the poisoned and clean images and their corresponding sampled texts for each of the three transformations, i.e., ΔAvg.ED (Backdoor − Clean).  The results are as follows:
>
> | Transformation | Description only | Font only | Language only |
> |---|---|---|---|
> | ΔAvg.ED (Backdoor − Clean) | 0.19817 | 0.25798 | 0.58463 |
>
>  In summary, the detection differences among the three individual transformations are caused by the differences in alignment between the poisoned images and the transformed text.
>
> * According to the following results, to reach near-max performance with the minimal set, the combinations of Font + Language and Language + Font are both good choices.
>
> | Attack |  BadNets |  |Blended |  |  WaNet |  | BadCLIP |   |
> |---|---|---| --|---| --|---| ---| --
> |  Transformation↓ | AUROC | F1-score |AUROC | F1-score |AUROC | F1-score | AUROC | F1-score |
> | Font+Description| 0.953 | 0.917 | 0.973 | 0.959 | 0.913 | 0.887 | 0.968| 0.961 |
> | Language+Font | 0.944 | 0.889 | 0.980 | 0.960 | 0.911 | 0.871 | 0.957| 0.951 |
> | Description+Language| 0.768 | 0.805 | 0.923 | 0.895 | 0.550 | 0.760 | 0.956|  0.910|
>
>
> **Q3: Thresholding is set via a clean reference set; sensitivity to the clean-set distribution and false-positive trade-offs is not fully explored.**
>
> R3: Thank you for the comment. To address this issue, using the BadCLIP attack as an example, we present the corresponding thresholds, along with FPR, calculated for the reference clean set with varying numbers of clean images, to analyze the sensitivity to the clean-set distribution and false-positive trade-offs. **As a result, as the number of clean reference images increases, the false positive rate (FPR) decreases.**
>
> | Number of clean reference images |threshold | FPR |
> |---|---|---|
> |10|35.4747|0.097|
> |20|35.5190|0.065|
> |30|35.5296|0.058|
> |40|35.5418|0.049|
> |50|35.6683|0.005|
>
> **Q4: Margins between clean and poisoned can be small, making the differences between shallow vs deep matching not as significant.**
>
> R4: We appreciate the reviewer’s insightful comment regarding the potential concern that the margins between clean and poisoned samples may be small, which could reduce the differences between shallow and deep matching. To investigate this,
> we report the differences (ΔAvg.ED) between the average Euclidean distance of clean images to the sampled texts and that of poisoned images to the sampled texts across attacks:
> * These results show that the margins are consistently substantial across attacks, indicating a clear distributional shift between poisoned and clean images.
> * Small intra-class variance for both clean and poisoned images shows that each class is compact and easily distinguishable.
>
> |Attack|Blended|Carlini\&Terzis|BadCLIP|
> |-|-|-|-|
> |ΔAvg.ED (Backdoor−Clean)|0.22818|0.21170| 0.22927|
> |Varience(Backdoor)|2.04929e-07|2.68294e-07|1.75753e-07|
> |Varience(Clean)|2.64913e-06|1.20742e-06|1.39513e-06|

---

> ### Author Response · Authors · 2025-11-23
>
> **Q5: Lack of theoretical supports for the shallow malicious matching vs deep benign matching.**
>
> R5: We thank the reviewer for this comment. Below we first give the intuition behind shallow malicious matching and deep benign matching, then we develop a theoretical model which formalizes this phenomenon and explains why it arises.
>
> **(1) Intuitive Explanation**
>
> * Deep Benign Matching: The robustness of matching for benign images **stems from CLIP's large-scale pre-training**. This process enables the model to learn rich visual concepts deeply connected to the text manifold of associated semantic concept. Consequently, the matching for clean images is robust to semantic-preserving text modifications.
> * Shallow Malicious Matching: In contrast, **backdoor learning forces a spurious correlation** between a trigger pattern and a specific target text, which often contradicts CLIP's foundational knowledge. This creates a fragile link where the poisoned image's malicious matching with the target text hardly generalizes to semantically equivalent variants of that text. Any minor modification to the target text can break this superficial connection.
>
> **(2) Theoretical Analysis**
>
> Let $\hat{f}_ v$ and $\hat{f}_ t$ be the backdoored CLIP image and text encoders, respectively, and denote image and text features by $\mathbf{z}_ v=\hat{f}_ v(\mathbf{v})$ and $\mathbf{z}_ t=\hat{f}_ t(\mathbf{t}) \in \mathbb{R}^d$. To formalize the intuition above, we first model the local text manifold corresponding to a semantic concept. For a concept $c$, we approximate the corresponding local text manifold of $c$ by the affine subspace:
>
> $S_ c:= \left\\{ \mathbf{\mu}_ c+U_ c \mathbf{\alpha}: \mathbf{\alpha} \in \mathbb{R}^k \right\\},$
>
> where $\mathbf{\mu}_c$ is the center, columns of $U_c \in \mathbb{R}^{d \times k}$ with $U_c^{\top} U_c=\mathbf{I}_k$ span this subspace.
>
>
> **Lemma 1.** Let $P_c:=U_c U_c^{\top}$ denote the projection matrix onto $S_c$. For any image feature $z_v$, its distance to the text manifold is
>
> $
> \operatorname{dis}\left(\mathbf{z}_v, S_c\right):=\min _\alpha\left\|\|\mathbf{z}_v-\left(\mathbf{\mu}_c+U_c \alpha\right)\right\|\|=\left\|\|\left(\mathbf{I}-P_c\right)\left(\mathbf{z}_v-\mathbf{\mu}_c\right)\right\|\| .
> $
>
> **Remark**: The term $(\mathbf{I} - P_c)(\mathbf{z}_v - \mathbf{\mu}_c)$ represents the residual part of the image feature that cannot be explained by the principal semantic directions of concept $c$. Thus, $\operatorname{dis}(\mathbf{z}_v, S_c)$ quantifies the semantic gap between the image and the concept manifold.
>
>
> **Proposition 1 (Existence of shallow malicious matching features).**  Let the local text manifold of the target concept $c^{\star}$ be approximated by an affine subspace $S_ {c^{\star}}=\left\\{\mathbf{\mu}_ {c^{\star}}+U_ {c^{\star}} \mathbf{\alpha}: \mathbf{\alpha} \in \mathbb{R}^k\right\\}$, where $\mathbf{\mu}_ {c^{\star}} \in \mathbb{R}^d$ and the columns of $U_ {c^{\star}} \in \mathbb{R}^{d \times k}$ form an orthonormal basis. Let the target text feature $\mathbf{z}_ t^{\star}=\hat{f}(\mathbf{t^{\star}}) \in \mathbb{R}^d$ lie on this manifold, i.e., $\mathbf{z}_ t^{\star} \in S_{c^{\star}}$, and denote its norm by $a:=\left\|\|\mathbf{z}_ t^{\star}\right\|\|>0$. Assume that for all benign images of their corresponding concept $c$, their image features $\mathbf{z}_ v^{\text {benign }}=\hat{f}(\mathbf{v}^{\text {benign}})$ satisfy $\operatorname{dis}\left(\mathbf{z}_ v^{\text {benign}}, S_ {c}\right) \leq \delta_ {\text {benign}}$ for some small $\delta_ {\text {benign }}>0$. For any $\varepsilon \in(0,1)$ satisfying
>
> $
> \delta_{\text {benign }}<a \sqrt{\frac{1}{(1-\varepsilon)^2}-1}, \quad (A)
> $
>
> there exists a feature vector $\mathbf{z}_v^{\text {poison}}=\hat{f}(\mathbf{v}^{\text{poison}}) \in \mathbb{R}^d$ such that
>
> $
> \cos \left(\mathbf{z}_ v^{\text {poison}}, \mathbf{z}_ t^{\star}\right) \geq 1-\varepsilon \quad \text {and} \quad \operatorname{dis}\left(\mathbf{z}_ v^{\text {poison}}, S_ {c^{\star}}\right)>\delta_ {\text {benign}}.
> $
>
> Moreover, if $\delta_ {\text {benign }}$ is sufficiently small compared to $a$, the distance $\operatorname{dis}\left(\mathbf{z}_ v^{\text {poison }}, S_ {c^{\star}}\right)$ can be larger than $\delta_ {\text {benign}}$, i.e., poisoned features can be almost perfectly aligned with the target text in cosine similarity while remaining far away from the target text manifold in the Euclidean distance, which is what we call shallow malicious matching.

---

> ### Author Response · Authors · 2025-11-23
>
> **Proof:** Since $\mathbf{z}_ t^{\star} \in S_{c^{\star}}$, there exists $\mathbf{\alpha}_ 0$ such that
>
> $
> \mathbf{z}_ t^{\star}=\mathbf{\mu}_ {c^{\star}}+U_ {c^{\star}} \mathbf{\alpha}_ 0 .
> $
>
>
> Equivalently, we have $\tilde{\mathbf{w}}:=\mathbf{z}_ t^{\star}-\mathbf{\mu}_ {c^{\star}} \in \operatorname{span}\left(U_ {c^{\star}}\right)$.
> Consider the orthogonal complement $\operatorname{span}\left(U_{c^{\star}}\right)^{\perp}$. For an arbitrary scalar $c>0$, choose any vector
>
> $
> \mathbf{u}_ {\perp} \in \operatorname{span}\left(U_ {c^{\star}}\right)^{\perp} \quad \text { with } \quad\left\|\|\mathbf{u}_ {\perp}\right\|\|=c .
> $
>
>
> We consider a simple construction of $\mathbf{z}_v^{\text{poison}}$:
>
> $
> \mathbf{z}_ v^{\text{poison}}:=\mathbf{z}_ t^{\star}+\mathbf{u}_ {\perp} .
> $
>
> **Cosine similarity with the target text.**
> Since $\left(\mathbf{z}_ t^{\star}\right)^{\top} \mathbf{u}_ {\perp}=0$, we have
>
> $
> \left\|\left\|\mathbf{z}_ v^{\text {poison }}\right\|\right\|^2=\left\|\left\|\mathbf{z}_ t^{\star}+\mathbf{u}_ {\perp}\right\|\right\|^2=\left\|\left\|\mathbf{z}_ t^{\star}\right\|\right\|^2+\left\|\left\|\mathbf{u}_ {\perp}\right\|\right\|^2=a^2+c^2,
> $
>
> and
>
> $
> \left(\mathbf{z}_ v^{\text {poison }}\right)^{\top} \mathbf{z}_ t^{\star}=\left(\mathbf{z}_ t^{\star}+\mathbf{u}_ {\perp}\right)^{\top} \mathbf{z}_ t^{\star}=\left\|\|\mathbf{z}_ t^{\star}\right\|\|^2=a^2 .
> $
>
>
> The cosine similarity between $\mathbf{z}_v^{\text {poison }}$ and $\mathbf{z}_t^{\star}$ is
>
> $
> \cos \left({\mathbf{z}}_v^{\text {poison}}, {\mathbf{z}}_t^{\star}\right)=\frac{\left(\mathbf{z}_v^{\text {poison}}\right)^{\top} \mathbf{z}_t^{\star}}{\left\|\left\|\mathbf{z}_v^{\text {poison}}\right\|\right\|\left\|\left\|\mathbf{z}_t^{\star}\right\|\right\|}=\frac{a^2}{\sqrt{a^2+c^2} a}=\frac{a}{\sqrt{a^2+c^2}}=\frac{1}{\sqrt{1+\frac{c^2}{a^2}}}.
> $
>
>
> Given any $\varepsilon \in(0,1)$, we want
>
> $
> \cos \left(\mathbf{z}_v^{\text {poison }}, \mathbf{z}_t^{\star}\right) \geq 1-\varepsilon,
> $
>
> this is equivalent to choose $c$ such that
>
> $
> c \leq a \sqrt{\frac{1}{(1-\varepsilon)^2}-1} .
> $
>
> **Distance to the text manifold.**
>
> By construction, we have
>
> $
> \mathbf{z}_ v^{\text {poison }}-\mathbf{\mu}_ {c^{\star}}=\left(\mathbf{z}_ t^{\star}-\mathbf{\mu}_ {c^{\star}}\right)+\mathbf{u}_ {\perp}=\tilde{\mathbf{w}}+\mathbf{u}_ {\perp}.
> $
>
> Since $\tilde{\mathbf{w}} \in \operatorname{span}\left(U_ {c^{\star}}\right)$ and $\mathbf{u}_ {\perp} \in \operatorname{span}\left(U_ {c^{\star}}\right)^{\perp}$, by the definition of $\operatorname{dis}\left(\cdot, S_ {c^{\star}}\right)$ in Lemma 1, we have
>
> $
> \operatorname{dis}\left(\mathbf{z}_ v^{\text {poison }}, S_ {c^{\star}}\right)=\left\|\left\|\left(I-U_ {c^{\star}} U_ {c^{\star}}^{\top}\right)\left(\mathbf{z}_ v^{\text {poison }}-\mathbf{\mu}_ {c^{\star}}\right)\right\|\right\|=\left\|\left\|\left(I-U_ {c^{\star}} U_ {c^{\star}}^{\top}\right)\left(\tilde{\mathbf{w}}+\mathbf{u}_ {\perp}\right)\right\|\right\|=\left\|\left\|\mathbf{0}+\mathbf{u}_ {\perp}\right\|\right\|=c .
> $
>
> Hence, if we choose $c>\delta_{\text {benign }}$, then
>
> $
> \operatorname{dis}\left(\mathbf{z}_ v^{\text {poison }}, S_ {c^{\star}}\right)=c>\delta_ {\text {benign }},
> $
>
> i.e., the poisoned image feature lies strictly farther away from the target manifold than any benign target-class image.
>
> **Combining the two constraints.**
> To simultaneously have
>
> $
> \operatorname{dis}\left(\mathbf{z}_ v^{\text {poison }}, S_ {c^{\star}}\right)=c>\delta_ {\text {benign }} \quad \text { and } \quad \cos \left(\mathbf{z}_ v^{\text {poison }}, \mathbf{z}_ t^{\star}\right) =\frac{1}{\sqrt{1+\frac{c^2}{a^2}}} \geq 1-\varepsilon,
> $
>
> we need a $c$ satisfying
>
> $
> \delta_{\text {benign }}<c \leq a \sqrt{\frac{1}{(1-\varepsilon)^2}-1} .
> $
>
>
> By assumption (A), such a $c$ exists. For any $c$ in this interval, picking any $\mathbf{u}_ {\perp}$ with $\left\|\left\|\mathbf{u}_ {\perp}\right\|\right\|=c$ in $\operatorname{span}\left(U_{c^{\star}}\right)^{\perp}$ and defining $\mathbf{z}_ v^{\text {poison}}= \mathbf{z}_ t^{\star}+\mathbf{u}_ {\perp}$ yields a feature that satisfies both inequalities.
>
> Moreover, if $\delta_{\text {benign }}$ is sufficiently small compared to $a$, the upper bound $a \sqrt{\frac{1}{(1-\varepsilon)^2}-1}$ is much larger than $\delta_{\text {benign}}$, so we can select $c$ with $c \gg \delta_{\text {benign}}$, making the gap $\operatorname{dis}\left(\mathbf{z}_ v^{\text {poison}}, S_ {c^{\star}}\right) / \delta_ {\text {benign}}$ large while keeping $\cos \left(\mathbf{z}_ v^{\text {poison}}, \mathbf{z}_ t^{\star}\right) \geq 1-\varepsilon$. $\square$

---

> ### Author Response · Authors · 2025-11-23
>
> **Remark:** in Proposition 1 we assume that for all benign images of their corresponding concept $c$, their image features $\mathbf{z}_ v^{\text {benign }}=\hat{f}\left(\mathbf{v}^{\text {benign }}\right)$ satisfy $\operatorname{dis}(\mathbf{z}_ v^{\text{benign}}, S_ {c}) \le \delta_ {\text{benign}}$ for some small $\delta_{\text {benign }}>0$. This assumption over deep benign matching naturally stems from CLIP's large-scale pre-training, where benign images of concept $c$ are well-aligned with a great number textual features $\left\\{\mathbf{z}_ t^{(j)}\right\\}_ {j=1}^m$ in $S_ {c}$ which approxiates the local text manifold of $c$. The goal for a benign image feature is to maximize the average cosine:
>
> $
> \frac{1}{m} \sum_ {j=1}^m \cos \left(\mathbf{z}_ v^{\text{benign}}, \mathbf{z}_ t^{(j)}\right)=\bar{\mathbf{\mu}}_ {c}^{\top} \bar{\mathbf{z}}_ v^{\text{benign}},
> $
>
> where $\bar{\mathbf{z}}_ v^{\text{benign}}={\mathbf{z}}_ v^{\text{benign}}/\|\|{\mathbf{z}}_ v^{\text{benign}}\|\|$, $\bar{\mathbf{\mu}}_ {c}=\frac{1}{m} \sum_ j \bar{\mathbf{z}}_ t^{(j)}/\left\|\|\bar{\mathbf{z}}_ t^{(j)}\right\|\| \in S_ {c}$. It's easy to find that The optimal $\bar{\mathbf{z}}_ v^{\text{benign}}$ is $\bar{\mathbf{\mu}}_ {c}/\|\|\bar{\mathbf{\mu}}_ {c}\|\|$, which lies in $S_ {c}$, thus its distance to $S_{c}$ is an extramely small $\delta_ {\text {benign }}$.
>
> In contrast, Proposition 1 shows that there exist backdoored features $z_v^{\text{poison}}$ which is aligned with the target caption in cosine similarity, while their distance to $S_ {c^{\star}}$ in the pre-normalization space is larger than $\delta_ {\text{benign}}$. This is what we called shallow malicious matching. Our Subspace Detection leverages precisely this geometric seperation to separate poisoned image features from deeply matched benign ones.
>
> **Q6: The method is not evaluated against label-consistent backdoor attacks or cases where poisoned images match multiple semantically similar target prompts. It is unclear whether the shallow-vs-deep matching assumption holds in these scenarios.**
>
> R6: Thanks for the insightful questions.
>
> (1) Evaluation against label-consistent backdoor attack: We have added an evaluation of Subspace Detection’s performance against label-consistent attack.
> * **Experimental setting:** We use a randomly generated 16x16 noise patch as the trigger to construct the  label-consistent attack. The patch is inserted into the center of the original images to produce the poisoned images. The evaluation is conducted on the ImageNet dataset.
> * **Experimental results:** Subspace Detection achieves an AUROC of 0.879 and an F1-score of 0.861 on the clean label attack.
> In summary, Subspace Detection remains effective even against label-consistent attacks.
>
> (2) Evaluation against multiple semantically similar target prompts:  We fully agree this is an insightful question. To address this concern, we designed an adaptive attack scenario to evaluate our method in Appendix F.2 of the paper.
> * **Experimental setting:**  We assumed the attacker was aware of all the description texts utilized for detection, and deployed the Blended attack using these description texts with target labels.
> * **Experimental results:**  Our method achieved an AUROC of 0.9602 and a F1-score of 0.9557. The results show that even when the attacker is aware of all the description texts used for detection, Subspace Detection still remains effective at detecting the poisoned images.

---

### Official Review · Reviewer_gpFW · 2025-11-01

**Soundness:** 3
**Presentation:** 3
**Contribution:** 2
**Rating:** 6
**Confidence:** 4

**Summary:**

This paper introduces a test time backdoor image detection method for CLIP pretrained with backdoor samples. The proposed method is based on the hypothesis that backdoor samples are further away from the region of composed of semantic variants of the target text, comparing to benign samples. Thus the proposed detection constructs subspaces with variants of the target text and compare the distance between the test sample and text features sampled from the subspace, to the max of such distances computed from a set of benign samples.

**Strengths:**

1. The paper is easy to follow, and the proposed detection is well motivated by empirical evidence and prior works.
2. The proposed detection method performs significantly better than baselines considered with respect to selected metrics. (although BDetCLIP is not evaluated, see Weaknesses)
3. A variety backdoor attacks are evaluated. (although clean label attacks is not evaluated, see Weaknesses)

**Weaknesses:**

1. minor: line 25 "board" -> "broad".
2. While the paper cites BDetCLIP as a test time backdoor detection method, it does not compare with it as a baseline.
3. I appreciate that the paper included a time complexity section. However, the section does not compare the detection overhead with CLIP inference time. Additionally, the subspace detection.
4. Based on Table 4, Language alone performs better than Language + Description. The conclusion "This indicates performance improvement mainly arises from the synergy among transformations rather than a single transformation alone" does not appear very obvious.

**Questions:**

1. How is the shallow matching of backdoors impacted by the poison rate and training dynamic like the number of training epochs.
2. How does the proposed analysis detection perform on clean label attack? I think this evaluation is especially important, since it could affect the distribution of the backdoor samples semantically.
3. How important is the diversity of the reference set with respect to the number of distinct semantic concepts?
4. How important is the ratio between the number of text variants sampled from each augmentation category?

---

> ### Author Response · Authors · 2025-11-22
>
> **Q1: minor: line 25 "board" -> "broad".**
>
> Thanks for pointing it out. We will modify it in the revision.
>
>
> **Q2: While the paper cites BDetCLIP as a test time backdoor detection method, it does not compare with it as a baseline.**
>
> R2: Thanks for the suggestion. We have added the BDetCLIP as a baseline in our experiments, as explained below:
>
> * **Experimental setting:** The evaluations are conducted on the validation sets of ImageNet-1K, ImageNet-R, and ImageNet-Sketch datasets under three backdoor attacks.
> * **Experimental results:** The detection results of our method demonstrate competitive performance with BDetCLIP, showing its effectiveness.
>
> | Detection          | Dataset→ | ImageNet-1K | ImageNet-R | ImageNet-Sketch |
> |--------------------|---------|------------- | ---------| --|
> |                    | Attack↓  | AUROC/F1-score | AUROC/F1-score | AUROC/F1-score |
> | BDetCLIP           | Blended | 0.957/0.915    | 0.813/0.748    | 0.727/0.755    |
> |                    | WaNet   | 0.950/ 0.909    | 0.702/0.685    | 0.704/0.713    |
> |                    | TrojVQA | 0.907/0.899    | 0.834/0.718    | 0.786/0.740    |
> |                    | **Average** | **0.938**/**0.908**    | **0.783**/**0.717**    | **0.739**/**0.736**    |
> | SubSpace Detection | Blended | 0.982/0.969    | 0.884/0.919    | 0.890/0.916    |
> |                    | WaNet   | 0.931/0.901    | 0.826/0.858    | 0.882/0.859    |
> |                    | TrojVQA | 0.925/0.879    | 0.865/0.856    | 0.863/0.854    |
> |                    | **Average** | **0.946**/**0.916**    | **0.858**/**0.878**    | **0.878**/**0.876**    |
>
>
> **Q3: I appreciate that the paper included a time complexity section. However, the section does not compare the detection overhead with CLIP inference time. Additionally, the subspace detection.**
>
> R3:
> (1) **Comparison with CLIP Inference Time**
>
> We thank the reviewer for this comment. A CLIP inference for ImageNet takes **0.00863** seconds per image. We will extend Table 10 with an additional “CLIP inference” column in the revised version.
>
> Our online detection cost of 0.009 seconds represents only an additional **4.1%** overhead relative to the standard inference time. This demonstrates that our method adds a minimal latency at test time, making it highly practical for real-world scenarios.
>
> (2) **Clarification on Subspace Detection Cost**
>
> For the reviewer's second comment, "Additionally, the subspace detection", we had provide the computational cost of Subspace Detection in Table 10. Here we would like to offer a more explicit clarification by mapping the specific algorithm steps to our reported timings.
>
> **The key design advantage of our method is that it naturally decomposes into a one-time, label-specific offline phase and a highly efficient online phase.**
>
>
> ***Offline Setup Cost***: This phase corresponds to the procedure in Section 4.2.1 "CONSTRUCTING A DISCRIMINATIVE SUBSPACE". The primary goal here is to pre-compute the mixture of Gaussians distribution for each class in the downstream dataset. The specific steps include:
> * Generating semantic text variants.
> * Approximating the local manifold by fitting a low-dimensional subspace.
> * Characterizing the region-of-interest within subspace by sampling along positive directions.
> * Fitting the uniform mixture of Gaussians to model this region.
>
> This offline phase takes about **0.1963 seconds per class**. Crucially, this distribution is label-specific and does not depend on any individual test image. Therefore, it can be pre-computed once per class and reused for all future test images predicted to belong to that class.
>
> ***Online Detection Cost:*** This phase corresponds to the procedure in Section 4.2.2 "DETECTING WITH THE SUBSPACE". This is the only computation that occurs at test-time for a single image's detection. The steps are:
> * Run CLIP inference to obtain its predicted label and the corresponding image feature. This step has complexity $O(\hat{f}_v+Kd)$, where $\hat{f}_v$ is the cost of the image encoder, $K$ is the number of classes, $d$ is feature dimension.
> * Load the pre-computed mixture of Gaussians distribution corresponding to the image's predicted class. This step has complexity $O(1)$.
> * Sample text features from this distribution. This step has complexity $O(n_sd)$, where $n_s$ is the number of samples.
> * Calculate their average Euclidean distance to the image feature. This step has complexity $O(n_sd)$
>
> The runtime of **0.009 seconds per image** reported in Table 10 refers to this efficient online detection phase. The overall per-image online complexity of our method is $O(\hat{f}_v+K d+n_sd)$
>
> We believe the above breakdown clarifies the test-time detection overhead for Subspace Detection. This decoupling is a core feature of our method's design, enabling it to be both highly effective and practically efficient.

---

> ### Author Response · Authors · 2025-11-22
>
> **Q4: Based on Table 4, Language alone performs better than Language + Description. The conclusion "This indicates performance improvement mainly arises from the synergy among transformations rather than a single transformation alone" does not appear very obvious.**
>
> R4: Thanks for the comments. We conduct additional experiments on BadNets to validate our conclusion. The results indicate that the detection performance of Language alone is not always superior to Language + Description across all attacks. We choose to explore combined transformations because we find that the worst-case performance of combined transformations is better than the worst-case performance of single transformation alone. This helps avoid the potential failure of the detection method due to the incorrect selection of a single transformation.
>
> | Attack→                   | BadNets |         |
> |-|-|-|
> | Transformation↓           | AUROC   | F1-score |
> | Language                  | 0.838   | 0.821    |
> | Font                      | 0.924   | 0.878    |
> | Description               | 0.749   | 0.802    |
> | Font+Description          | 0.953   | 0.917    |
> | Language+Font             | 0.944   | 0.889    |
> | Description+Language      | 0.768   | 0.805    |
> |All| 0.962   | 0.920    |
>
> Q5: How is the shallow matching of backdoors impacted by the poison rate and training dynamic like the number of training epochs.
>
> **R5: Thanks for the comment. We conduct BadCLIP attack at different poisoning rates and training epochs to investigate their effects on the shallow matching of backdoors.**
>
> * **Epochs**: In BadCLIP attacks, poisoned images are paired with natural descriptions containing the target label. As the number of training epochs increases, the model progressively strengthens the alignment between the poisoned images and their target class descriptions, which can lead to a slight decrease in the detection performance of Subspace Detection. However, when constructing the text manifold, we use multiple descriptions that do not contain the target label, along with transformations like Language and Font. This diversity in the text manifold makes it difficult for the attacker to comprehensively cover all the varied semantic forms, allowing Subspace Detection to maintain effectiveness.
>
> | Epoch | AUROC | F1-score |
> |---|---|---|
> | 5 | 0.995 | 0.976 |
> | 10 | 0.966 | 0.963 |
> | 15 | 0.962 | 0.931 |
> | 20 | 0.957 | 0.907 |
> * **Poisoning rate**: Although an increase in the number of poisoned training images helps strengthen the alignment between poisoned images and the target label, Subspace Detection constructs a text manifold using various transformations such as Description, Font, and Language. As a result, poisoned images cannot fully align with the diverse text features within the manifold, ensuring that Subspace Detection remains effective.
>
> | Poisoning rate | AUROC | F1-score |
> |---|---|---|
> | 0.01 | 0.966 | 0.963 |
> | 0.03 | 0.961      |  0.958     |
> | 0.05 |  0.959     |  0.955     |
> | 0.07 |  0.955     |  0.954     |
>
> **In summary, changing the poisoning rate and training epochs can not improve the shallow matching of backdoors. Subspace Detection maintains its effectiveness across a range of poisoning rates and training epoch settings.**

---

> ### Author Response · Authors · 2025-11-22
>
> **Q6: How does the proposed analysis detection perform on clean label attack? I think this evaluation is especially important, since it could affect the distribution of the backdoor samples semantically.**
>
> R6: Thanks for the insightful question. We have added an evaluation of Subspace Detection’s performance against clean label attacks.
> * **Experimental setting:** We use a randomly generated 16x16 noise patch as the trigger to construct the clean label attack. The patch is inserted into the center of the original images to produce the poisoned images. The evaluation is conducted on the ImageNet dataset.
> * **Experimental results:** Subspace Detection achieves an AUROC of 0.879 and an F1-score of 0.861 on the clean label attack.
> In summary, Subspace Detection remains effective even against clean label attacks.
>
>
>
> **Q7: How important is the diversity of the reference set with respect to the number of distinct semantic concepts?**
>
> R7: Thanks for the insightful comment. We completely agree with the importance of considering the diversity of the reference set with respect to the number of distinct semantic concepts. This is also the reason why we chose AUROC as our evaluation metric. AUROC measures the overall performance of the detection method across all possible thresholds, providing a comprehensive view of its effectiveness. By using AUROC, we avoid any dependency on a single threshold, ensuring that the model's performance is evaluated in a more holistic manner, incorporating all possible variations in the reference set. Based on the results provided in the paper, the performance of our detection method is stable and effective.
>
>
> **Q8: How important is the ratio between the number of text variants sampled from each augmentation category?**
>
> Q8: Thanks for the comment. We increase 10 sampled text variants for each transformation to adjust the ratio between the number of text variants sampled from the Description, Font, and Language categories. The analysis is based on the BadCLIP attack. The results show that although the detection effectiveness improves with an increase in the number of sampled text variants, the improvement is most pronounced for the Language transformation, followed by the Font transformation.
>
> | Transformation | AUROC | F1-score |
> |---|---|---|
> | Description | 0.968 | 0.947 |
> | Font | 0.971 | 0.965 |
> | Language | 0.979 | 0.967 |

---

### Official Review · Reviewer_qkGi · 2025-11-01

**Soundness:** 2
**Presentation:** 3
**Contribution:** 2
**Rating:** 4
**Confidence:** 3

**Summary:**

This paper investigates the vulnerability of backdoored CLIP models and identifies a phenomenon termed shallow malicious matching. Building on this insight, the authors propose Subspace Detection, a test-time poisoned sample detection method that constructs a low-dimensional subspace of the CLIP text manifold using semantically varied text transformations to distinguish benign from poisoned samples. Experiments demonstrate its effectiveness.

**Strengths:**

1. This paper is original in its formulation of the shallow malicious matching phenomenon and its introduction of a subspace-based detection framework for identifying test-time poisoned samples in multimodal models like CLIP.
2. This paper is well-structured, with clear motivation.

**Weaknesses:**

1. Lack related baselines. There are some works focusing on test-time detection, such as [1].
2. The motivation underlying the mixture of Gussian model is unclear. The author should give detailed formulation to explain.
3. In ablation study of time $L$, the performance does not reach the peak at $L=4$. More results on a larger value of $L$ should be reported.
4. The adopted distance metic, $d_2$, needs more consideration and discussion.



[1] Test-Time Multimodal Backdoor Detection by Contrastive Prompting.

**Questions:**

See weaknesses.

---

> ### Author Response · Authors · 2025-11-22
>
> **Q1: Lack related baselines. There are some works focusing on test-time detection, such as [1].**
>
> [1] Test-Time Multimodal Backdoor Detection by Contrastive Prompting.
>
> R1: Thanks for the suggestion. We have added the BDetCLIP[1] as a baseline in our experiments, as explained below:
>
> * **Experimental setting:** The evaluations are conducted on the validation sets of ImageNet-1K, ImageNet-R, and ImageNet-Sketch datasets under three backdoor attacks.
> * **Experimental results:** The detection results of our method demonstrate competitive performance with BDetCLIP, showing its effectiveness.
>
> | Detection          | Dataset→ | ImageNet-1K | ImageNet-R | ImageNet-Sketch |
> |--------------------|---------|------------- | ---------| --|
> |                    | Attack↓ | AUROC/F1-score | AUROC/F1-score | AUROC/F1-score |
> | BDetCLIP           | Blended | 0.957/0.915    | 0.813/0.748    | 0.727/0.755    |
> |                    | WaNet   | 0.950/ 0.909    | 0.702/0.685    | 0.704/0.713    |
> |                    | TrojVQA | 0.907/0.899    | 0.834/0.718    | 0.786/0.740    |
> |                    | **Average** | **0.938**/**0.908**    | **0.783**/**0.717**    | **0.739**/**0.736**    |
> | SubSpace Detection | Blended | 0.982/0.969    | 0.884/0.919    | 0.890/0.916    |
> |                    | WaNet   | 0.931/0.901    | 0.826/0.858    | 0.882/0.859    |
> |                    | TrojVQA | 0.925/0.879    | 0.865/0.856    | 0.863/0.854    |
> |                    | **Average** | **0.946**/**0.916**    | **0.858**/**0.878**    | **0.878**/**0.876**    |

---

> ### Author Response · Authors · 2025-11-22
>
> **Q2: The motivation underlying the mixture of Gaussian model is unclear. The author should give detailed formulation to explain.**
>
> R2: We thank the reviewer for this comment. Our motivation underlying the mixture of Gussian model is twofold, stemming from both theoretical grounding and empirical validation, which we will elaborate on below.
>
> **(1) Theoretical Motivation**
>
> The core of our detection method is to determine whether a test image is poisoned based on its feature's relationship with the corresponding text manifold. This can be formally expressed within a Bayesian inference framework.
>
> Let's first define the key variables:
>
> ${y}_d$: A binary variable indicating if a test image ${\mathbf{v}}$ is poisoned (${y}_d=1$) or clean (${y}_d=0$).
>
> ${\mathbf{z}}_v$: The feature vector of the test image ${\mathbf{v}}$.
>
> ${\mathbf{z}}_t$: The feature vector of the original text caption predicted for ${\mathbf{v}}$.
>
> ${\mathbf{z}}_d$: A text feature vector sampled from the region-of-interest on the text manifold, which is used for detection.
>
> * **A Bayesian inference framework**
>
> Our goal is to compute the posterior probability $p({y}_d|{\mathbf{z}}_v, {\mathbf{z}}_t)$. We can express the detection distribution as:
>
> $p({y}_d|{\mathbf{z}}_v, {\mathbf{z}}_t) =\int p({y}_d|{\mathbf{z}}_v,{\mathbf{z}}_d)p({\mathbf{z}}_d| {\mathbf{z}}_t)d {\mathbf{z}}_d, \quad (1)$
>
> where $p({y}_d|{\mathbf{z}}_v,{\mathbf{z}}_d)$ is the likelihood of an image being poisoned given its feature ${\mathbf{z}}_v$ and a specific text feature ${\mathbf{z}}_d$ used for detection. We model this using a threshold-based detector as detailed in Sec. 4.2.2:
>
> $p\left(y_d=1 \mid \mathbf{z}_v, \mathbf{z}_d\right)=\mathbb{I}\left(\left\|\mathbf{z}_v-\mathbf{z}_d\right\|_2>\tau^\prime\right).$
>
> The second part $p({\mathbf{z}}_d| {\mathbf{z}}_t)$ represents the distribution of discriminative text features conditioned on the original predicted text feature. The core of our method lies in constructing a good model for this distribution.
>
> * **A Single Gaussian**
>
> As detailed in Sec. 4.2.1 of our paper, instead of sampling within the entire local text manifold's subspace approximation $S$, our method captures a region-of-interest within it. This region is designed to contain text features that maximally amplify the separation between benign and poisoned images. Therefore, we use the distribution of features within this region to model $p(\mathbf{z}_d| \mathbf{z}_t)$.
>
> To characterize this region, we begin with a set of handcrafted text variant features and additional features sampled along the positive direction inside $S$. Let $\mathbf{\mu}_ {\text{bias}}$ and $\mathbf{D}_ \text{bias}$ are the mean and the deviation matrix of this feature set. All these features within this region should satisfy two key properties:
>
> (a) They are centered around the mean of the defining feature set.
>
> (b) They remain constrained within the subspace $S$ and should not deviate into irrelevant parts of the feature space.
>
> This constructive process can be formally expressed by representing $\mathbf{z}_ d$ from the region as: $\mathbf{z}_ d = \mathbf{\mu}_ {\text{bias}} + \eta\mathbf{D}_ \text{bias}\mathbf{\phi}$, where $\mathbf{\phi} \sim \mathcal{N}(\mathbf{0}, \mathbf{I})$ is a vector of standard normal random variables and $\eta$ is a scaling constant. **This formulation directly induces a Gaussian distribution for the features in the region-of-interest:**
>
> $p= \mathcal{N}({\mathbf{\mu}}_ {\text{bias}},\eta^2\mathbf{D}_ \text{bias}\mathbf{D}_ \text{bias}^\top).$
>
> Note that the set of handcrafted text variant features and additional sampled text features are generated from the original text $\mathbf{z}_ t$, making this Gaussian a valid model for $p(\mathbf{z}_ d| \mathbf{z}_ t)$.

---

> ### Author Response · Authors · 2025-11-22
>
> * **A Mixture of Gaussians**
>
> **However, a single Gaussian model is a strong and often inaccurate assumption for representing the complex, potentially multi-modal geometry of a local text manifold. A Mixture of Gaussians provides a more powerful and flexible model, capable of capturing a richer, more accurate approximation of the true distribution.** The process of sampling text variants to define our region-of-interest introduces stochasticity, meaning a single constructed $p$ is just one approximation of the true underlying manifold region. To create a more robust and accurate representation, we repeat the subspace construction process (i.e., sampling, filtering, and modeling) $L$ times. Each iteration $l \in \\{1, ..., L\\}$ yields a slightly different Gaussian distribution $\mathcal{N}({\mathbf{\mu}}_ {\text{bias}}^{(l)},\eta^2\mathbf{D}_ \text{bias}^{(l)}\mathbf{D}_ \text{bias}^{(l)\top})$. By modeling the overall distribution as a uniform mixture of these $L$ Gaussians, we obtain a more powerful and flexible model:
>
> $p_ {mix}= \frac{1}{L}\sum_ {l=1}^L\mathcal{N}({\mathbf{\mu}}_ {\text{bias}}^{(l)},\eta^2\mathbf{D}_ \text{bias}^{(l)}\mathbf{D}_ \text{bias}^{(l)\top}).$
>
> This Mixture of Gaussians could provide a much more stable and reliable distribution $p({\mathbf{z}}_ d| {\mathbf{z}}_ t)$ for the Bayesian inference in Eq. (1).
>
> **(2) Empirical Validation**
>
> This theoretical motivation is strongly supported by our ablation studies. In Section 5.3 Impact of region-of-interest modeling times $L$ of our paper, we empirically investigate how the number of Gaussian components ($L$) affects detection performance. As shown in Figure 4, as we increase $L$, the detection performance consistently improves. This result empirically validates that a Mixture of Gaussians ($L>1$) provides a more accurate and effective characterization of the discriminative region on the text manifold compared to a single Gaussian ($L=1$).
>
> In summary, our use of a Mixture of Gaussians is motivated by the need for a flexible and robust model of the discriminative text feature distribution, which is theoretically grounded in Bayesian inference and empirically validated by our ablation studies. We thank the reviewer again for prompting this clarification. We also refine the motivation in the main text.

---

> ### Author Response · Authors · 2025-11-22
>
> **Q3: In ablation study of time $L$, the performance does not reach the peak at $L=4$. More results on a larger value of $L$ should be reported.**
>
> R3: We appreciate the reviewer’s suggestion. Following your suggestion, we have extended the ablation study to larger values of $L$(from $L=1$ to $L=7$), and the full results are shown in below Table. From these results, we observe when $L$ increases from 1 to 4, the performance improves steadily across attacks. This confirms that multiple modeling iterations of the region-of-interest, which more appropriately characterize this region, are indeed beneficial. As $L$ further increases from 4 to 7, the detection performance becomes very stable, and no significant improvements is observed. Since the computational cost for characterizing the region-of-interest grows with $L$, we choose $L=3$ in all main experiments as a trade-off between cost and performance.
>
>
> |Attack| BadNet |          | Blended |          | SIG   |          | WaNet ||TrojVQA || Carlini\&Terzis |          | BadCLIP |          |
> |-|-|-|-|-|-|----------|-------|----------|---------|----------|-----------------|----------|-|-|
> | $L$    | AUROC  | F1-score | AUROC   | F1-score | AUROC | F1-score | AUROC | F1-score | AUROC   | F1-score | AUROC           | F1-score | AUROC   | F1-score |
> | 1      | 0.929  | 0.878    | 0.966   | 0.927    | 0.668 | 0.753    | 0.902 | 0.851    | 0.875   | 0.821    | 0.983           | 0.936    | 0.960   | 0.933    |
> | 2      | 0.954  | 0.908    | 0.983   | 0.958    | 0.693 | 0.769    | 0.927 | 0.895    | 0.909   | 0.863    | 0.991           | 0.959    | 0.967   | 0.958    |
> | 3      | 0.962  | 0.920    | 0.982   | 0.969    | 0.692 | 0.788    | 0.931 | 0.901    | 0.925   | 0.879    | 0.994           | 0.968    | 0.966   | 0.963    |
> | 4      | 0.961  | 0.924    | 0.983   | 0.974    | 0.698 | 0.791    | 0.941 | 0.912    | 0.926   | 0.879    | 0.995           | 0.972    | 0.965   | 0.965    |
> | 5      | 0.963  | 0.931    | 0.982   | 0.968    | 0.694 | 0.787    | 0.930 | 0.907    | 0.924   | 0.876    | 0.996           | 0.948    | 0.965   | 0.961    |
> | 6      | 0.962  | 0.925    | 0.981   | 0.971    | 0.693 | 0.789    | 0.933 | 0.906    | 0.923   | 0.881    | 0.995           | 0.956    | 0.964   | 0.960    |
> | 7      | 0.963  | 0.927    | 0.983   | 0.973    | 0.694 | 0.787    | 0.932 | 0.901    | 0.924   | 0.880    | 0.994           | 0.965    | 0.966   | 0.963    |

---

> ### Author Response · Authors · 2025-11-22
>
> **Q4: The adopted distance metic, $d_2$, needs more consideration and discussion.**
>
> R4: We thank the reviewer for this comment. In our method, $d_2(\mathbf{u}, \mathbf{v})=\|\|\mathbf{u}-\mathbf{v}\|\|_2$ denotes the **Euclidean distance**. We will clarify it in the revision. Below we discuss how Euclidean distance is motivated.
>
> (1) **Consistency with Our PCA subspace:**
> In our method, we employ PCA to obtain a low-dimensional subspace to approximate the local text manifold. A fundamental property of PCA is that it finds the projection that minimizes the reconstruction error under the $L_2$ norm. Consequently, our linear approximation of the manifold is, by construction, optimal in an $L_2$ sense. By subsequently using the Euclidean distance to measure the deviation of an image feature from this PCA-constructed manifold, we ensure (i) the construction of the subspace and (ii) the deviation measure used for detection are based on the same underlying geometry, avoiding a mismatch between how the manifold is learned and how distances to it are measured.
>
> (2) **Consistency with Our Gaussian Modeling:**
> In Sec. 4.2.1, we model the text features $\mathbf{z}_d$ in the discriminative region-of-interest in local text manifold, $p(\mathbf{z}_d|\mathbf{z}_t)$, using Gaussian distributions. This assumption implies a local Gaussian structure for text feature variations. Given that:
> * Clean images are well-aligned with the region-of-interest in the CLIP embedding space, the feature of a clean image ($\mathbf{z}_v$) should also adhere to this local geometry. The difference between the image feature and a text feature on the manifold ($\mathbf{z}_d$) can be modeled as a small perturbation. For a clean image, this residual, $\mathbf{\delta} = \mathbf{z}_v - \mathbf{z}_d$, can be viewed as approximately Gaussian noise around the local text manifold, i.e., $\mathbf{\delta} \sim \mathcal{N}(\mathbf{0}, \sigma^2 \mathbf{I})$.
> * Poisoned image features either have a much larger variance or a bias that pushes them away from the same region.
>
> The negative log-likelihood of $\mathbf{\delta}$ is then directly proportional to $\|\|\mathbf{\delta}\|\|_2^2$, which is the squared Euclidean distance. Therefore, thresholding $d_2\left(\mathbf{z}_v, \mathbf{z}_d\right)$ is equivalent to using a sufficient statistic for distinguishing high-likelihood (near-manifold, i.e., clean) samples from low-likelihood (off-manifold, i.e., poisoned) samples. Our detector based on $d_2\left(\mathbf{z}_v, \mathbf{z}_d\right)$ can thus be interpreted as a non-parametric approximation of a likelihood-ratio test under this local Gaussian assumption.
>
> (3) **Experimental Comparison with $L_1$ Distance**:
>
> | Attack | BadNet |  | Blended |  | SIG |  | WaNet |  | TrojVQA |  | Carlini\&Terzis |  | BadCLIP |  |
> |---|---|---|---|---|---|---|---|---|---|---|---|---|---|---|
> | Metric | AUROC | F1-score | AUROC | F1-score | AUROC | F1-score | AUROC | F1-score | AUROC | F1-score | AUROC | F1-score | AUROC | F1-score |
> | $L_1$ | 0.671 | 0.695 | 0.693 | 0.681 | 0.525 | 0.533 | 0.872 | 0.923 | 0.738 | 0.827 | 0.512 | 0.497 | 0.490 | 0.532 |
> | $L_2$ | 0.962 | 0.920 | 0.982 | 0.969 | 0.692 | 0.788 | 0.931 | 0.901 | 0.925 | 0.879 | 0.994 | 0.968 | 0.966 | 0.963 |
>
> In above table, we experiment with replacing the $L_2$ distance by $L_1$ in our detection method, i.e., using $\|\left\|\mathbf{z}_v-\mathbf{z}_d\right\|\|_1$ instead of $\left\|\|\mathbf{z}_v-\mathbf{z}_d\right\|\|_2$. This modification consistently led to noticeably worse detection performance across multiple attacks. This is because $L_1$ is no longer aligned with (i) the PCA subspace construction and (ii) the Gaussian-based modeling of region-of-interest discussed above.

---

### Comment · Area_Chair_GFLP · 2025-11-27

Dear Reviewers,

The discussion phase will end soon. The authors have provided responses. Please take a look and see if your concerns are addressed via official comments.

Thanks for your efforts and contributions to ICLR 2026.

Best regards,

Your Area Chair

---

### Author Response · Authors · 2025-12-01

Dear AC,

Below we provide a brief summary of our work and our responses to the reviewers' comments.


### **Summary of Our Work**

* **Task**: We propose **a test-time detection method, Subspace Detection,** to detect poisoned images in backdoored CLIP models.

* **Motivation and Novelty**: We observe that while benign images exhibit deep alignment with the semantic text manifold of their captions (i.e., **deep benign matching**), poisoned images show only a superficial, shallow alignment with the target text (i.e., **shallow malicious matching**).

* **Contribution**: Leveraging this insight, our method distinguishes poisoned from clean images by approximating the local text manifold, identifying a discriminative region, and computing the Euclidean distance between the test image feature and text features within this region. A large distance indicates a poisoned image. Experimental results demonstrate that our method significantly outperforms prior detection methods and maintains strong detection capability against a range of state-of-the-art backdoor attacks and across multiple downstream datasets.

### **Recognized Strengths**

* **Novelty and Originality**: R-qkGi and R-PUEo recognized the **originality** of the insight. R-qkGi and R-gpFW noted our method was **well-motivated**. R-PUEo noted the **method is conceptually clear and methodologically coherent**.

* **Clarity of Algorithm:** R-gpFW, R-PUEo and R-Zp8v noted the method was **easy to follow and increased practical applicability**.

* **Writing**: R-qkGi and R-PUEo noted the paper was **well-written and structured**.

### **Shared Reviews**

*   **Comparison with BDetCLIP (all Reviewers):** We clarified the fundamental difference between the methods (in Q1 of R-PUEo). BDetCLIP relies on the **insensitivity** of poisoned images to semantic changes. In contrast, our method exploits the **fragility** of poisoned images, failing to generalize to semantic-preserving textual changes. Besides, a comparison with the BDetCLIP detection method has been added to the original manuscript, demonstrating that our method outperforms BDetCLIP.

*   **Theoretical Support for Shallow vs. Deep Matching (R-Zp8v, R-PUEo):** We provided both intuitive and formal explanations to elaborate on the mechanism behind shallow malicious matching. We prove a proposition showing there exists features that have high cosine similarity to the target text (as backdoored images do) yet have a much larger Euclidean distance to the manifold than benign images.

*   **Ablation study on the impact of individual text transformation (R-qkGi, R-Zp8v):** We showed the performance of our method on additional attacks under various transformation combinations, further confirming the necessity of multiple transformations.

### **Negative Reviews**

* **Detailed response to R-qkGi (Score: 4):**

    * lack of baseline BDetCLIP: we added the BDetCLIP comparison, showing our method's superiority.

    * motivations for mixture of Gaussians and Euclidean distance: we provided both theoretical and experimental clarifications for these designs.

    * ablation of parameter $L$: we extended the ablation study on $L$ to verify our parameter choice.

* **Detailed response to R-Zp8v (Score: 4):**

    * lack of baseline BDetCLIP: we added the BDetCLIP comparison, showing our method's superiority.

    * analysis of single: transformation differences-We computed the difference in average Euclidean distance between poisoned and clean images and their corresponding sampled texts for each of the three transformations, i.e., ΔAvg.ED(Backdoor−Clean), demonstrating that detection differences were caused by the differences in alignment between the poisoned images and the transformed text.

    * sensitivity to the clean-set distribution and false-positive trade-offs: We presented thresholds and corresponding FPRs calculated on reference clean sets of varying sizes to analyze sensitivity to the clean-set distribution and false-positive trade-offs.

    * concern regarding small margins between clean and poisoned samples: We provided the difference in average Euclidean distance between poisoned and clean images and their corresponding sampled texts for each of the three transformations, i.e., ΔAvg.ED(Backdoor−Clean), to demonstrate that the margins were not small. Besides, small intra-class variance for both clean and poisoned images showed that each class is compact and easily distinguishable.

    * lack of theoretical supports for the shallow malicious matching vs deep benign matching: We provided both intuitive and formal explanations to elaborate on the mechanism behind shallow malicious matching. We prove a proposition showing there exists features that have high cosine similarity to the target text (as backdoored images do) yet have a much larger Euclidean distance to the manifold than benign images.

Thank you for your time and consideration.

Best regards,

The Authors

---

### Meta-Review · Area_Chair_FmGX · 2026-01-06

**Summary:**

The reviewers have the following concerns in their original reviews:
- insufficient comparison with the state-of-the-art baseline BDetCLIP
- lack of theoretical grounding
- inadequate ablation studies, particularly on the impact of the parameter L and the choice of distance metric
- limited evaluation against advanced or clean-label backdoor attacks and other multimodal models
- questions about the robustness of threshold selection, sensitivity to the clean-set distribution, and the practical computational overhead relative to standard CLIP inference.

The authors' rebuttal has addressed most of the key concerns, so I recommend Accept.

**Reviewer Concerns:**

The authors have addressed most of the key concerns in the rebuttal. They added comprehensive comparisons with BDetCLIP, provided both intuitive and formal theoretical support for shallow vs. deep matching and the Gaussian modeling, extended ablation studies, and included new experiments on clean-label attacks, additional advanced attacks, and other models like OpenCLIP.

**Reviewer Scores:**

Reviewer qkGi (Score: 4): Likely to raise the score to 6 (above threshold) due to the added baseline, extended ablations, and theoretical clarifications.

Reviewer gpFW (Score: 6): Might keep the positive score, given the thorough responses, especially on baseline comparison, clean-label evaluation, and overhead analysis.

Reviewer Zp8v (Score: 4): Would likely raise the score to 6 (above threshold) as major concerns about baselines, theoretical support, and transformation analysis were well addressed.

Reviewer PUEo (Score: 6): Might keep the positive score, given the detailed methodological comparison with BDetCLIP and additional experiments on other models and attacks.

---

### Decision · Program_Chairs · 2026-01-26

Accept (Poster)